# Hydrogeochemical Characterization of Groundwater at the Boundaries of Three Aquifers in Central México

Guadalupe Ibarra-Olivares [1,2], Raúl Miranda-Avilés [3,*], José A. Ramos-Leal [4], Janete Morán-Ramirez [4], María Jesús Puy-Alquiza [3], Yanmei Li [3], Edgar Ángeles-Moreno [3] and Pooja Kshirsagar [3]

1   Doctoral Program of Water Science and Technology, Engineering Division, University of Guanajuato, Guanajuato 36000, Mexico; g.ibarraolivares@ugto.mx
2   Mexican Geological Survey, Pachuca 42080, Mexico
3   Department of Mining, Metallurgy and Geology, University of Guanajuato, Guanajuato 36020, Mexico
4   Potosine Institute for Scientific and Technological Research, Applied Geosciences Division, San Luis Potosí 78216, Mexico
*   Correspondence: rmiranda@ugto.mx

**Abstract:** This study investigates the natural hydrogeochemical mechanisms that govern groundwater chemistry at the margins of the Silao-Romita, Valle de León, and La Muralla aquifers in Mexico's "Bajío Guanajuatense". The wells of the La Muralla aquifer have temperatures ranging from 25 to 45 °C, while in the valleys, the temperatures range from 25 to 29 °C. In the Sierra de Guanajuato recharge zone, the thermal spring registers 95 °C. High Na concentrations (125 to 178 mg/L) are measured due to thermalism. One sample includes 316 mg/L of $SO_4$, which is related to a potential gypsum zone. Three hydrogeochemical facies are identified (Ca-Mg $HCO_3$, Na-Ca-$HCO_3$, and Na-$HCO_3$). The hydrogeochemical characterization and processes imply hydraulic linkage via regional thermal flows enhanced by faults and the mixing of local flow waters with intermediate flows. The isotopic results indicate that part of the groundwater volume has been exposed to local evaporation processes due to the presence of surface water bodies and irrigation returns. The highest isotopic enrichment is observed near or in the recharge regions. In contrast, the most depleted zones are in the valleys, where there is a more significant interaction with the rock and a longer residence time, implying a mixture of local water flows with deeper or intermediate flows, which, when combined with water geochemistry, indicates a connection between the aquifers studied. The Kruskal–Wallis variance tests, used to compare the differences between aquifers, show that the Valle de León aquifer has more significant differences with respect to the Silao-Romita and La Muralla aquifers. This study's findings are essential for one of central Mexico's most populous and economically active areas.

**Keywords:** aquifer interaction; geological faults; hydrogeochemistry processes; Guanajuato

## 1. Introduction

About 25% of all water that is withdrawn for irrigation and 50% that is used for household purposes worldwide comes from groundwater resources [1]. They are also essential water sources for human use, primarily supplying agricultural irrigation needs, among other purposes [2–5]. Despite this resource's great value, it is underutilized, ineffectively managed, and even overexploited [6–8].

Groundwater becomes the primary, and occasionally the only, source of supplies for the population, agriculture, and industry in arid and semi-arid regions in central and Northern Mexico, mainly where precipitation is less than 500 mm/year. Considering that water that is pumped at greater depths is mineralized mostly because of the residence time and interaction with geological materials, it is essential to characterize groundwater to detect hydrogeochemical processes, changes in its chemical composition, and the mixing of different groundwater sources via local or regional flows [9,10]. Arsenic and fluoride

have been reported in some areas of Bajío Guanajuatense [11–13]. The origin, relationship to geological features, or hydraulic connections between aquifers have only been explored in a few rare instances despite reports of high amounts of arsenic in groundwater used by more than 1.5 million people [14].

Some works have characterized low-temperature thermal systems in central Mexico or water–rock interaction processes using hydrogeochemistry [15–17] and using a multivariate statistical analysis of geochemical data to determine the processes that control the geochemical evolution of groundwater [18,19]. On the other hand, the authors of [9] investigated the impact of geological faults on thermalism and the levels of fluoride and arsenic in the groundwater of the Laguna Seca aquifer in Guanajuato State.

Even though most authors discuss the significances of geological faults and fractures in the chemical composition of water [20,21], they do not perform structural geological investigations or establish a well-sampling criterion based on the traces of regional lineaments and geological faults, as is suggested in the current study.

In this work, we hypothesize that there are hydrogeochemical processes that evidence the interrelation or mixing between the groundwater of the fractured aquifer (La Muralla) and the granular aquifers (Valle de León and Silao-Romita). Therefore, the main objective of this research is to identify the natural hydrogeochemical processes that control groundwater chemistry, considering lithological units, structural geology, and groundwater flow directions at the boundaries of the Silao-Romita, the Valle de León, and the La Muralla aquifers in the "Bajío Guanajuatense", Mexico. Identifying hydrogeochemical facies and flow types will provide valuable information regarding the hydraulic connection between aquifers and their relationships with geologic conditions.

## 2. Materials and Methods

### 2.1. Study Area

The study area is in the central-western portion of the State of Guanajuato in Silao, Romita, San Francisco del Rincón, and the León de los Aldama Municipality. It covers an area of 2500 km$^2$. The study area partially includes three aquifers: the southwestern part of the Silao-Romita aquifer, the southern part of the Valle de León aquifer, and the La Muralla aquifer. It is bounded to the northeast and east by the Sierra de Guanajuato, to the southwest and south by the Sierras de Pénjamo and El Veinte, as well as by small hills to the northwest, where the La Muralla aquifer is located. The surface drainage drains from the north to south through two main streams, the Silao River and the Guanajuato River, both of which converge in the southern part of the study area (Figure 1).

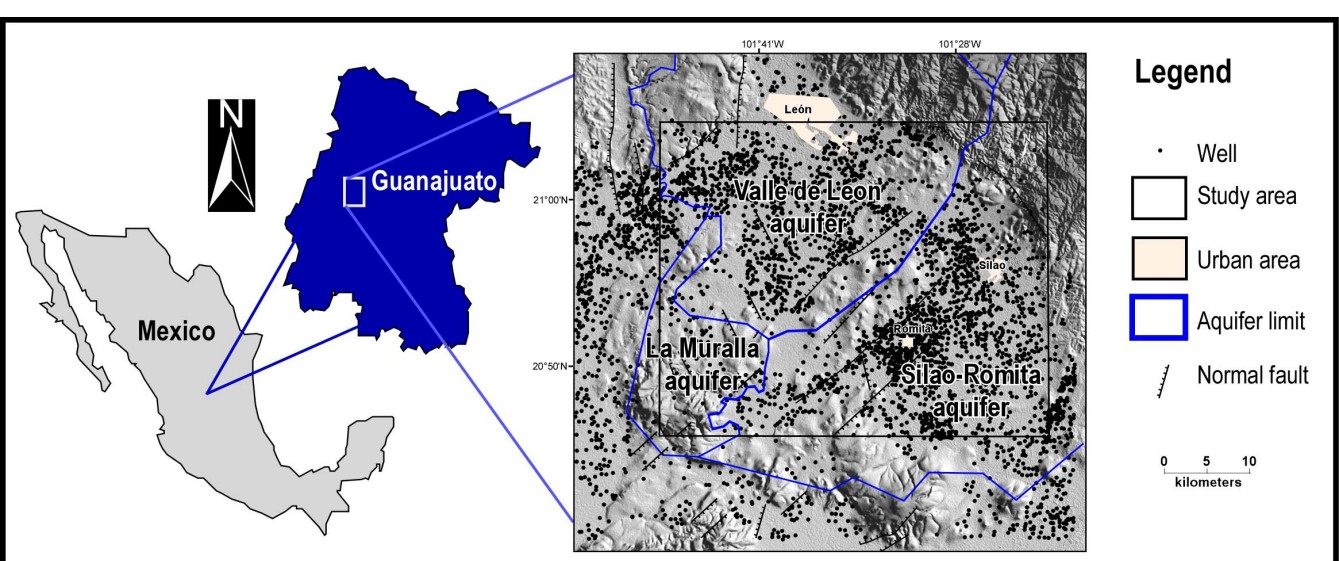

**Figure 1.** Location of the study area: boundary of the Silao-Romita, La Muralla, and Valle de León aquifers.

Physiographically, the northern portion of the study area is in the Mesa Central province, while the southern part corresponds to the physiographic province of the Neovolcanic Axis [22]. The Mesa Central province, also known as the Mexican Altiplano, is affected by the volcanism of the Sierra Madre Occidental. Vast plains interrupted by scattered mountain ranges are evident, among them being the Sierra de Guanajuato, whose southwestern flank constitutes the physiographic discontinuity called Valles Paralelos, comprising rocks of diverse lithology and abrupt topography. The physiographic discontinuity of parallel valleys in the southwestern Sierra de Guanajuato is characterized by the vertical convex shapes of its valleys [23].

The Silao-Romita and Valle de León aquifers extract 333 and 121 $Mm^3$/year, respectively, according to the Guanajuato State Water Commission (CEAG, Guanajuato, México). The amount of groundwater extraction from the La Muralla aquifer is equal to 29 $Mm^3$/year [24]. In locations with irrigation returns, infiltration accounts for 20 to 40% of the total irrigation use; however, due to the depth of the piezometric level, only 10% is anticipated to reach the aquifer [24]. The area covered by aquifers is mostly temperate subhumid, according to Köppen's classification [23].

The highest average temperatures are registered in the Río Turbio and León valleys, with temperatures between 18 and 19 °C, and particularly in the Silao area, with temperatures of up to 20 °C. In the topographically high areas, such as Sierra de Guanajuato and Cerro Grande, the lowest temperatures occur between 14 and 16 °C. Similarly, the highest rainfall is recorded in these topographic eminences with up to 800 mm values. The lowest rainfall occurs in the Turbio River and Silao valleys, ranging from 600 to 650 mm. Sierra de Guanajuato has the lowest amounts of evapotranspiration, computed by the Lturc technique, at 450 mm. The highest values are 590 mm in Cerro Grande, located in the northern portion of the study area. Due to meteorological and geological characteristics, Sierra de Guanajuato is a critical recharge zone for the Valle de León and Silao-Romita aquifers.

*2.2. Geology*

The physiographic region of the Neovolcanic Axis includes the León, Silao-Romita, and La Muralla aquifers [25]. The mountainous regions comprise Oligocene tuffs and lavas, with Miocene–Pliocene lavas in the highest parts. Most of the surface under study comprises valleys, where there is Quaternary alluvium up to 50 m thick and lacustrine sedimentary fill of the Miocene age between 150 and 250 m thick [24,26].

Lithological contacts were taken from the cartography of the National Institute of Statistics and Geography (INEGI, Aguascalientes, Mexico), Guanajuato State Water Commission (CEAG, Guanajuato, México, 2001), and corrected with field observations. The Jurassic–Cretaceous rocks of the regional geologic basement are in Sierra de Guanajuato [25,27]. However, these rocks have not been detected in the studied area, so the Eocene Guanajuato Conglomerate is considered the hydrogeologic basement, with an average thickness of 1500 m. The Guanajuato Conglomerate comprises conglomerates, sedimentary breccias, and sandstones [28].

The aquifers studied are part of a tectonic half-graben known as the El Bajío continental basin [25,29], bounded to the east by the NW-SE El Bajío fault system. The El Bajío basin is segmented by NE-SW normal faults, wholly or partially covered by recent sediments (Figure 2).

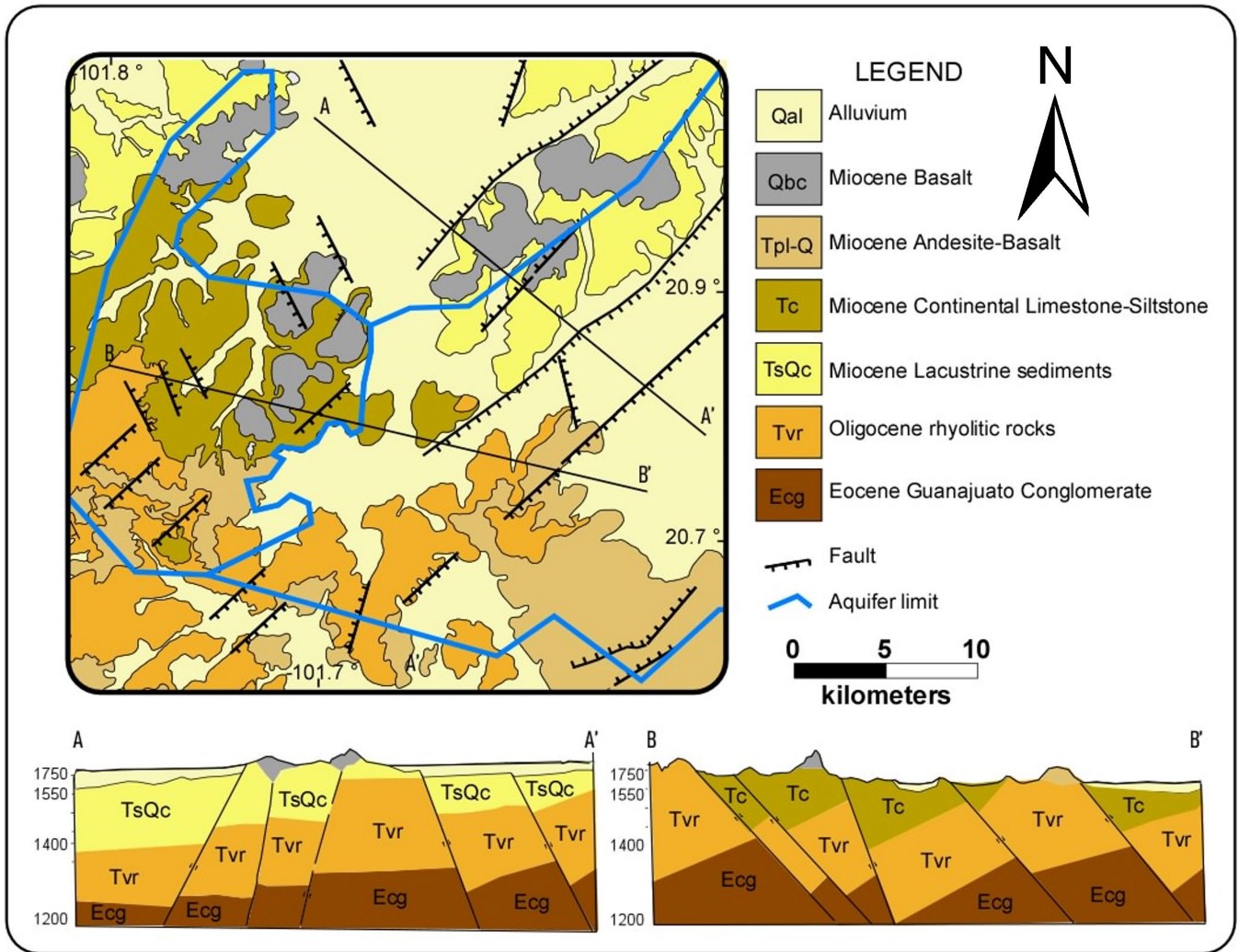

**Figure 2.** Geologic map and profiles of the study area: boundary of the Silao-Romita, La Muralla, and Valle de León aquifers.

### 2.3. Hydrogeology

The La Muralla aquifer is located west of the study area and covers 241 km$^2$. It is bounded to the northeast by the Valle de León aquifer and to the east by the Silao-Romita aquifer. It is a free to semi-confined aquifer in a fractured setting, with piezometric depths ranging from 60 to 150 m. It consists of a 120 m thick alluvial matrix and rhyolitic rocks. Transmissivity values range from 0.08 to 70.3 × 10$^{-3}$ m$^2$/s. Groundwater is important since it serves 78% of the León, Gto municipality.

The Valle de León and Silao-Romita aquifers are located north and southeast of the study area, respectively, and they are bordered on the west by the La Muralla aquifer. Both are free to semi-confined aquifers, with a surface size of 707 km$^2$ for Valle de León and 1881 km$^2$ for Silao-Romita. They are primarily found in granular alluvial and lacustrine materials with an average saturated thickness of 200 m in the top section and rhyolitic rocks in the lower part. Silao-Romita has a transmissivity of 4.0 × 10$^{-6}$ to 58.0 × 10$^{-3}$ m$^2$/s and a conductivity of 0.03 to 10.0 m/d (3.5 × 10$^{-7}$ to 1.16 × 10$^{-4}$ m/s) [23]. It is important to note that more than 75% of the groundwater in both aquifers is used for agricultural irrigation.

### 2.4. Inventory of Groundwater Exploitations

We analyzed information on the location, construction, and operation of 1375 active wells in the Silao-Romita aquifer, 126 in the La Muralla aquifer, and 1188 in the Valle de León aquifer [26,30] to observe the distribution of wells and the boundaries between aquifers

(Figure 3). Based on the piezometric history, a monitoring campaign was programmed to verify the current groundwater levels and corroborate the congruence of the historical measurements of 22 previous years.

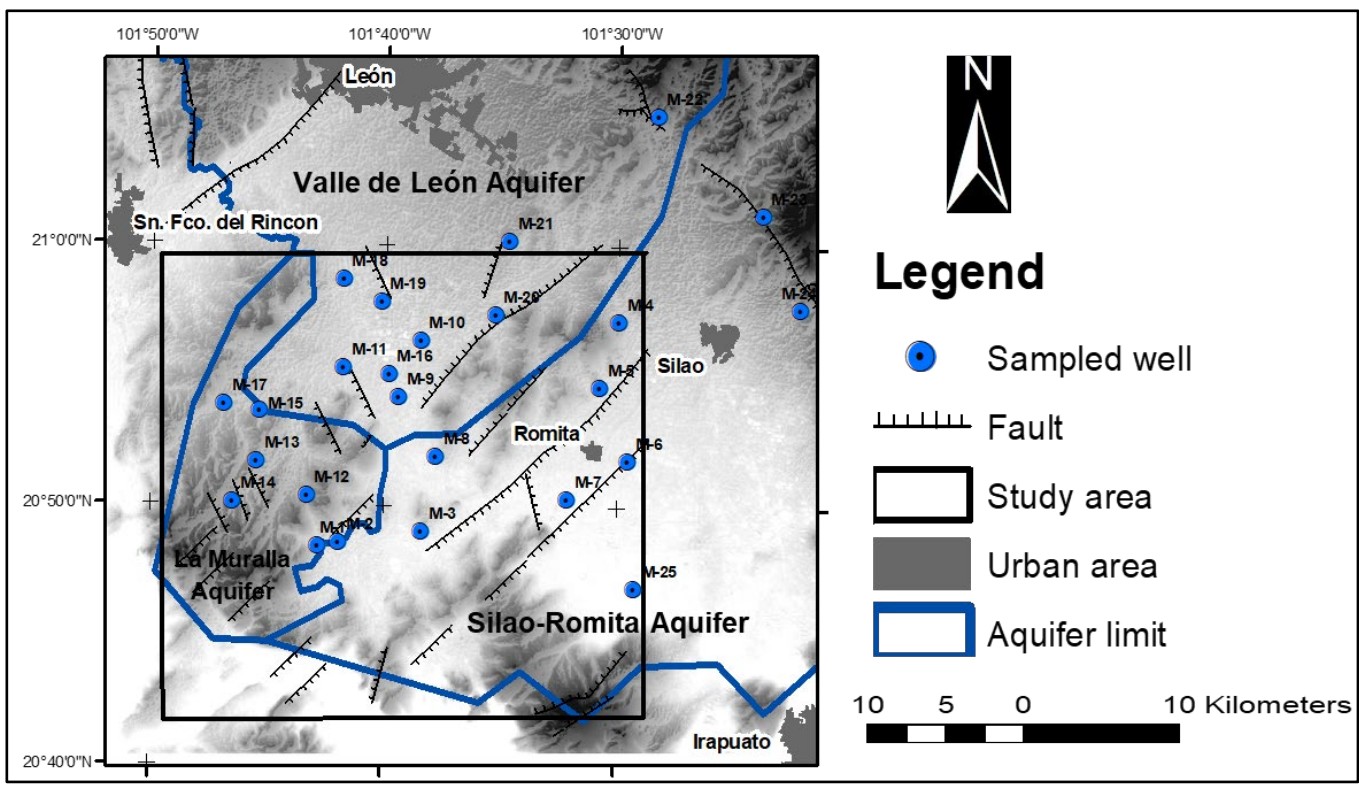

**Figure 3.** Location of sampled well in the study area: boundary of the Silao-Romita, La Muralla, and Valle de León aquifers.

### 2.5. Groundwater Quality Analysis and Sampling Program

Information from the water quality networks of the La Muralla, Silao-Romita, and Valle de León aquifers was analyzed. The monitoring networks of the three aquifers comprise 64 wells, but only 14 are of interest. The sampling sites were chosen based on the following criteria: (a) wells located at the boundaries of the granular and fractured aquifers, (b) deep wells with a piezometric level representative of the study area, (c) wells with a piezometric history, (d) wells located in fault and thermal zones, and (e) wells located in recharge zones. The chosen period was a rainfall period in 2022, based on an analysis of CEAG data from three years before. The physicochemical analyses of these monitoring networks include temperature, conductivity, and Total Dissolved Solids. A total of 24 deep wells and one thermal spring were sampled for physicochemical analysis; the Surface and Groundwater Sampling Protocol for the Analysis of Heavy Metals, Major Ions, and Isotopes EIA-D001 of the Mexican Geological Survey was used for the field sampling of the groundwater wells, as well as the Official Mexican Standard NOM-014-SSA1, 1993 [31]. A HANNA multiparameter, model HI9829, was used to determine the electrical conductivity (EC), temperature, and pH. The anions in the water were determined via ion, chromatography ASTM (2011). Standard test method for anions in water by suppressed ion chromatography ASTM D4327-11 [32]. The monitoring program was carried out at the boundaries of the three aquifers and the main recharge zone (Sierra de Guanajuato), as shown in Figure 3. Each site visited was georeferenced using a GPS with coordinates—WGS-84 datum—and piezometric levels were measured with a previously calibrated electrical probe. In accordance with NOM-014-SSA1-1993, a water sample was collected at each sampling station for the physicochemical analysis, both for anions and cations, with the cation sample acidified to identify heavy metals. We used the Arc Map

10.5 software to undertake a spatial distribution analysis of the main physicochemical parameters using the Kriging interpolation method.

*2.6. Isotopic Analysis*

For the Oxygen 18 and Deuterium isotopic analyses, the water samples were sent to the Stable Isotope Laboratory of the Department of Environmental and Soil Sciences of the National Laboratory of Geochemistry and Mineralogy of the Institute of Geology of the UNAM. The water samples were filtered with Titan 3 filters, with a pore size of 0.20 μm and a 30 mm diameter PES (polyethylene sulfone) membrane, and placed in 2.0 mL glass vials (approximately 1.75 mL, sample with Teflon septa). Isotopic measurements of the water samples were performed using a Liquid Water Isotope Analyzer (LWIA) Los Gatos Research Inc. (LGR, Mountain View, CA, USA) connected to an LC PAL liquid auto-injector (908-0008-9001, CTC Analytics AG, Zwingen, Switzerland). The results of $\delta^{18}O$ and $\delta^2H$ VSMOW (Vienna Standard Mean Ocean Water) were normalized using VSMOW and VSLAP (Standard Light Antarctic Precipitation), according to Coplen, 1988. For the isotopic analysis of $\delta^{18}O$ and $\delta^2 H$, the LWIA (Liquid Water Isotope Analyzer) equipment was used with laser absorption technology called OA-ICOS (Off-Axis Integrated Cavity Output Spectroscopy), which is based on measuring the energy differences of hydrogen and oxygen.

*2.7. Statistical Data Analysis*

Groundwater data were summarized using simple descriptive statistics. Pearson's correlation (r) was utilized to examine the link between groundwater's physical and chemical constituents. The three aquifers under investigation and their physiochemical parameters were compared using a non-parametric variance test (Kruskal–Wallis). This method compares numerous independent random samples and can be used as a non-parametric alternative to one-way ANOVA [33,34].

**3. Results and Discussion**

*3.1. Hydrogeochemistry*

The physicochemical parameters of the groundwater are presented in Table 1. The pH of the samples in the study area is alkaline in nature, ranging from 7 to 9, with an average of 7.59 (Table 2) The Electrical Conductivity (EC) values vary between 340 and 1023 μS/cm, with an average of 564 μS/cm. The Total Dissolved Solids (TDS) values range from 217 to 654 mg/L. Seven wells have temperatures above 30 °C, and the thermal spring (95 °C) has a significantly lower water quality due to considerable increases in sodium and sulfates. The main cations in decreasing order are Na > Ca > Mg > K. Sodium is the most abundant cation, ranging from 24 to 148 mg/L. The calcium values vary from 1 to 139 mg/L. The magnesium and potassium values are relatively low, ranging from 0.06 to 19 mg/L for magnesium and from 1 to 15 mg/L for potassium. The anions show abundance in the following order: $HCO_3$ > $SO_4$ > Cl. Bicarbonate is the dominant ion in groundwater, with its concentration ranging from 232 to 375 mg/L. The sulfates have values between 6 and 316 mg/L. Chloride has values between 1 and 9 mg/L. The hydrogeochemical parameters exhibit substantial variability, suggesting that complex hydrogeochemical processes occur in the study area.

**Table 1.** Field parameters and concentrations of major ions at the boundaries of the Leon Valley, Silao Romita, and La Muralla aquifers.

| ID | Coordinates | | Physico-Chemical Parameters | | | | Cations (mg/L) | | | | Anions (mg/L) | | |
|---|---|---|---|---|---|---|---|---|---|---|---|---|---|
| | Latitude | Longitude | PH | T °C | EC (µS/cm) | TDS (mg/L) | $Ca^{2+}$ | $Mg^{2+}$ | $Na^+$ | $K^+$ | $HCO_3^-$ | $Cl^-$ | $SO_4^{2-}$ |
| M-1 | 20°48′24″ | 101°42′47″ | 7.55 | 30.50 | 502.00 | 321.28 | 36.67 | 5.49 | 79.12 | 4.04 | 290.00 | 3.51 | 51.77 |
| M-2 | 20°48′34″ | 101°41′53″ | 7.56 | 29.10 | 547.00 | 350.08 | 46.49 | 9.42 | 67.93 | 5.43 | 319.00 | 3.34 | 35.28 |
| M-3 | 20°49′01″ | 101°38′20″ | 7.61 | 28.70 | 384.00 | 245.76 | 31.08 | 5.67 | 75.70 | 6.00 | 298.00 | 2.86 | 36.36 |
| M-4 | 20°57′07″ | 101°29′58″ | 7.58 | 25.00 | 402.00 | 257.28 | 39.76 | 9.01 | 29.26 | 15.80 | 262.00 | 7.70 | 13.14 |
| M-5 | 20°54′36″ | 101°30′46″ | 7.61 | 25.80 | 491.00 | 314.24 | 27.33 | 6.49 | 72.07 | 12.99 | 302.00 | 2.36 | 22.63 |
| M-6 | 20°51′47″ | 101°29′30″ | 7.34 | 23.80 | 695.00 | 444.80 | 76.40 | 30.34 | 41.68 | 3.88 | 375.00 | 9.68 | 46.25 |
| M-7 | 20°50′17″ | 101°32′07″ | 7.51 | 30.90 | 671.00 | 429.44 | 29.75 | 6.90 | 125.78 | 6.01 | 296.00 | 5.28 | 135.83 |
| M-8 | 20°51′54″ | 101°37′46″ | 7.28 | 25.00 | 1023.00 | 654.72 | 139.01 | 1.52 | 93.24 | 3.63 | 304.00 | 8.97 | 316.01 |
| M-9 | 20°54′10″ | 101°39′22″ | 7.29 | 21.40 | 517.00 | 330.88 | 57.43 | 11.84 | 35.33 | 9.57 | 300.00 | 4.28 | 31.23 |
| M-10 | 20°56′21″ | 101°38′27″ | 7.70 | 24.50 | 340.00 | 217.60 | 38.60 | 7.17 | 24.83 | 6.27 | 232.00 | 1.94 | 6.28 |
| M-11 | 20°55′17″ | 101°41′46″ | 7.58 | 38.60 | 484.00 | 309.76 | 49.09 | 2.09 | 48.30 | 5.76 | 206.00 | 2.48 | 94.66 |
| M-12 | 20°50′ 22″ | 101°43′18″ | 7.49 | 31.40 | 614.00 | 392.96 | 40.62 | 7.82 | 86.45 | 5.16 | 298.00 | 6.55 | 87.21 |
| M-13 | 20°51′38″ | 101°45′27″ | 7.28 | 30.40 | 428.00 | 273.92 | 41.61 | 6.45 | 44.15 | 7.23 | 302.00 | 1.33 | 7.79 |
| M-14 | 20°50′04″ | 101°46′27″ | 9.03 | 41.30 | 502.00 | 321.28 | 1.49 | 0.06 | 117.34 | 1.24 | 292.00 | 2.13 | 11.56 |
| M-15 | 20°53′34″ | 101°45′20″ | 7.46 | 28.70 | 641.00 | 410.24 | 44.44 | 6.12 | 89.98 | 6.42 | 305.00 | 7.75 | 102.84 |
| M-16 | 20°55′01″ | 101°39′48″ | 7.60 | 19.50 | 449.00 | 287.36 | 43.21 | 14.02 | 33.63 | 7.85 | 295.00 | 1.87 | 13.31 |
| M-17 | 20°53′50″ | 101°46′51″ | 7.49 | 35.90 | 497.00 | 318.08 | 35.31 | 8.57 | 59.39 | 7.81 | 298.00 | 3.45 | 15.04 |
| M-18 | 20°58′38″ | 101°41′48″ | 7.47 | 26.20 | 455.00 | 291.20 | 40.88 | 19.63 | 29.14 | 8.00 | 306.00 | 2.94 | 13.69 |
| M-19 | 20°57′47″ | 101°40′08″ | 7.47 | 23.60 | 414.00 | 264.96 | 39.56 | 14.86 | 30.27 | 7.57 | 299.00 | 1.76 | 11.02 |
| M-20 | 20°57′21″ | 101°35′15″ | 7.36 | 25.30 | 524.00 | 335.36 | 52.90 | 9.07 | 48.56 | 9.73 | 308.00 | 5.08 | 28.65 |
| M-21 | 21°00′11″ | 101°34′42″ | 7.60 | 26.20 | 402.00 | 257.28 | 45.78 | 8.98 | 25.66 | 10.10 | 278.00 | 1.71 | 8.16 |
| M-22 | 21°05′01″ | 101°28′22″ | 8.71 | 95.00 | 743.00 | 475.52 | 2.37 | 0.05 | 148.23 | 7.88 | 310.00 | 16.38 | 27.96 |
| M-23 | 21°01′16″ | 101°23′48″ | 7.53 | 21.80 | 973.00 | 622.72 | 28.43 | 23.59 | 178.78 | 6.95 | 605.00 | 16.10 | 14.26 |
| M-24 | 20°57′41″ | 101°22′09″ | 7.17 | 26.20 | 765.00 | 489.60 | 76.11 | 34.94 | 42.02 | 6.67 | 453.00 | 11.21 | 22.69 |
| M-25 | 20°46′54″ | 101°29′10″ | 7.56 | 25.90 | 633.00 | 405.12 | 56.43 | 19.48 | 54.38 | 7.83 | 308.00 | 6.43 | 52.41 |

**Table 2.** Descriptive statistics of physicochemical parameters in groundwater.

| | N | Range | Minimum | Maximum | Mean | Std. Deviation |
|---|---|---|---|---|---|---|
| PH | 25 | 1.86 | 7.17 | 9.03 | 7.5932 | 0.40795 |
| T °C | 25 | 75.50 | 19.50 | 95.00 | 30.4280 | 14.39793 |
| EC | 25 | 683.00 | 340.00 | 1023.00 | 563.8400 | 173.37019 |
| TDS | 25 | 437.00 | 218.00 | 655.00 | 360.8400 | 111.05730 |
| Ca | 25 | 137.52 | 1.49 | 139.01 | 44.8300 | 25.97602 |
| Mg | 25 | 34.89 | 0.05 | 34.94 | 10.7832 | 8.82665 |
| Na | 25 | 153.95 | 24.83 | 178.78 | 67.2488 | 40.53269 |
| K | 25 | 14.56 | 1.24 | 15.80 | 7.1928 | 2.98369 |
| $HCO_3$ | 25 | 399.00 | 206.00 | 605.00 | 313.6400 | 74.63460 |
| Cl | 25 | 15.05 | 1.33 | 16.38 | 5.4836 | 4.25592 |
| $SO_4$ | 25 | 309.73 | 6.28 | 316.01 | 48.2412 | 65.31975 |

### 3.2. Spatial Distribution of Major Ions

A spatial distribution analysis was performed to determine the distribution of Total Dissolved Solids (TDS) concentrations, major ions, temperature, depth, and flow rates in wells extracting groundwater. The hydrogeochemical parameters show a high variation in their concentrations, indicating that complex hydrogeochemical processes occur in the study area. The ion concentrations increase towards the east. Many ions with high concentrations are located northeast and southeast of the study area. The Total Dissolved Solids (TDS) show maximum values in the northeast and south-central regions due to the water–rock interaction and the direction of subsurface flow from north to south. The spatial variation of the TDS diagram shows generally good groundwater quality in the northwest and southwest zones (Figure 4). Physiographically, these areas correspond to valleys for the Valle de Leon granular aquifer and correspond to gentle slopes for the fractured La Muralla aquifer. Gentle slopes can increase the level of dissolved ions due to the longer contact time and the water–rock interaction. On the other hand, the semi-arid climate is also

an important parameter contributing to the high concentration of TDS [15,35,36]. On the other hand, in the central-northern and central-eastern zones of the research area, granular materials (alluvial and lacustrine) constitute the lithological units, and basalts are found in the structural highs in the central area. Finally, underlying the recent materials, rhyolitic rocks and lacustrine materials outcrop in the south-central and west-central portions. These geological units may be related to the high concentrations of calcium, magnesium, sodium, bicarbonates, and sulfates in the groundwater.

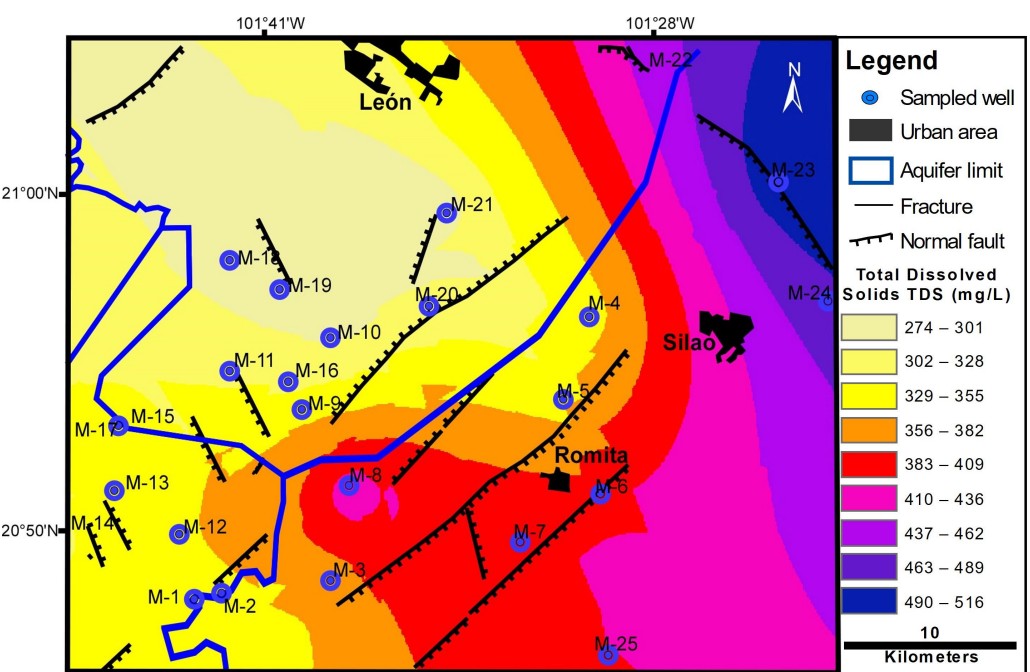

**Figure 4.** Spatial variation of Total Dissolved Solids (TDS).

The sodium spatial variation diagram in Figure 5a shows the same trend as the TDS; its concentrations fluctuate between 24 and 178 mg/L. The interaction between water and volcanic rocks generates the presence of sodium. The calcium in Figure 5b presents 1 to 139 mg/L concentrations. It shows a high concentration in the central portion of the area, which indicates the interaction with calcareous lacustrine rocks in the areas surrounding La Muralla. The magnesium in Figure 5c could be associated with the dissolution of dolomite, one of the main minerals in carbonate rocks. However, it is found in the area in low concentrations of 2 to 35 mg/L. The potassium concentrations are very low, from 1 to 15 mg/L, and its origin may be associated with the clays found in the alluvial deposits (Figure 5d). Compared to the high Na contents, the K deficiency may be attributed to the high weathering resistance of potassium and the fixation of clay minerals [19].

Bicarbonate is distributed in the lower zones, valley zones, and areas with abatement cones, or areas with a higher density of wells, as shown in Figure 6a, except for one sample that shows high concentration values (600 mg/L) in a well that is located northeast of the study area. The groundwater in these zones has recently been infiltrated and mixed to the west with thermal water (35 to 45 °C) from the La Muralla aquifer. Its bicarbonate concentrations fluctuate between 232 and 605 mg/L, averaging 313 mg/L. This concentration results from a natural geological process in which the recharge water interacts with the various fill materials that comprise the strata's most superficial layers. The sulfate ion is generally present in low concentrations in the zone, and only in two wells located in the central area, at the limits of the aquifers, do its concentrations increase. The values fluctuate between 6 and 316 mg/L, with an average of 48 mg/L, as shown in Figure 6b. They are produced naturally because of the leaching of gypsum-containing rocks and sulfate-containing soils. Chloride ion (Cl⁻) is present in low concentrations

between 1 and 16 mg/L. The highest concentration occurs in an anomalous well, where high concentrations of sodium are also observed, suggesting a natural origin (Figure 6c).

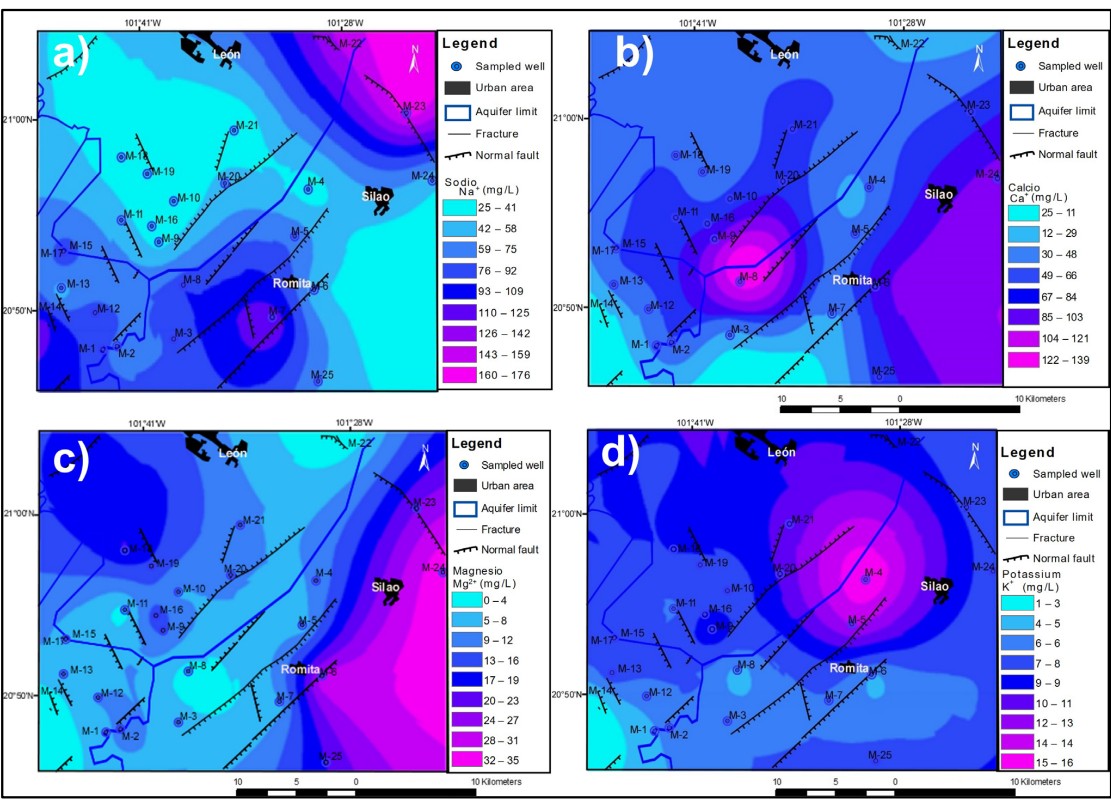

**Figure 5.** Spatial variation of cations (mg/L): (**a**) Na$^+$, (**b**) Ca$^+$, (**c**) Mg$^{2+}$, and (**d**) K$^+$.

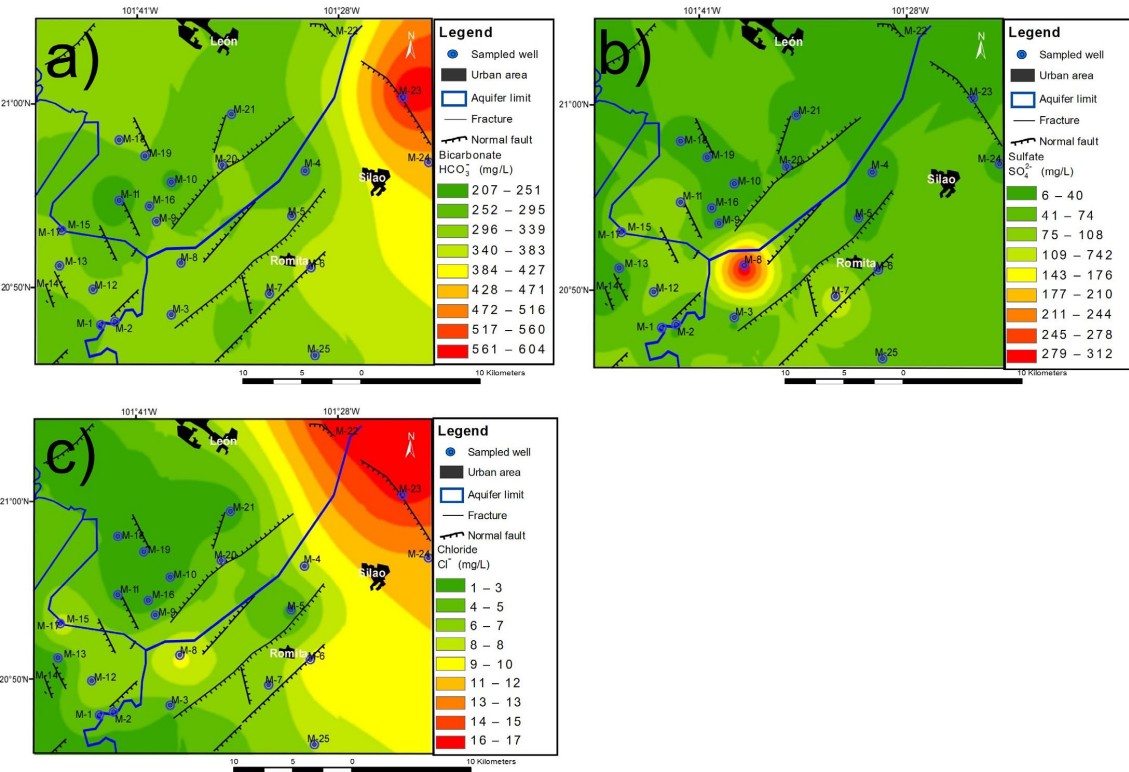

**Figure 6.** Spatial variation of anions (mg/L): (**a**) HCO$_3^-$, (**b**) SO$_4^{2-}$, and (**c**) Cl$^-$.

### 3.3. Groundwater Types

Figure 7 shows a Piper diagram where three hydrogeochemical facies were identified: calcium–magnesium bicarbonate ($HCO_3$-Ca-Mg), calcium–sodium bicarbonate ($HCO_3$-Ca-Na), and sodium bicarbonate ($HCO_3$-Na). Six samples correspond to thermal wells with temperatures of 30 to 35 °C, which shows a process of cation exchange and water–rock interaction [37]. In specific cases where the temperatures were higher than 45 °C, the composition increased notably in the sodium and potassium due to being in fault or fracture zones. The hydrogeochemical facies, $HCO_3$-Ca-Mg, correspond to a younger or recently infiltrated water type. In contrast, the $HCO_3$-Ca-Na and $HCO_3$-Na facies have longer residence times and more considerable contact with the rocks, resulting in much more mature waters. According to the Stiff diagrams and the geology of the study area (Figure 8), variations or mixtures of water are observed in the central and southwestern zones of the study area. Most of the ions present high concentrations of bicarbonates in the low or valley zones (north and west). In contrast, major ions are mixed in the higher zones (center and southwest), with sodium, phosphates, and bicarbonates being predominant. High concentrations of sodium are observed in thermal wells.

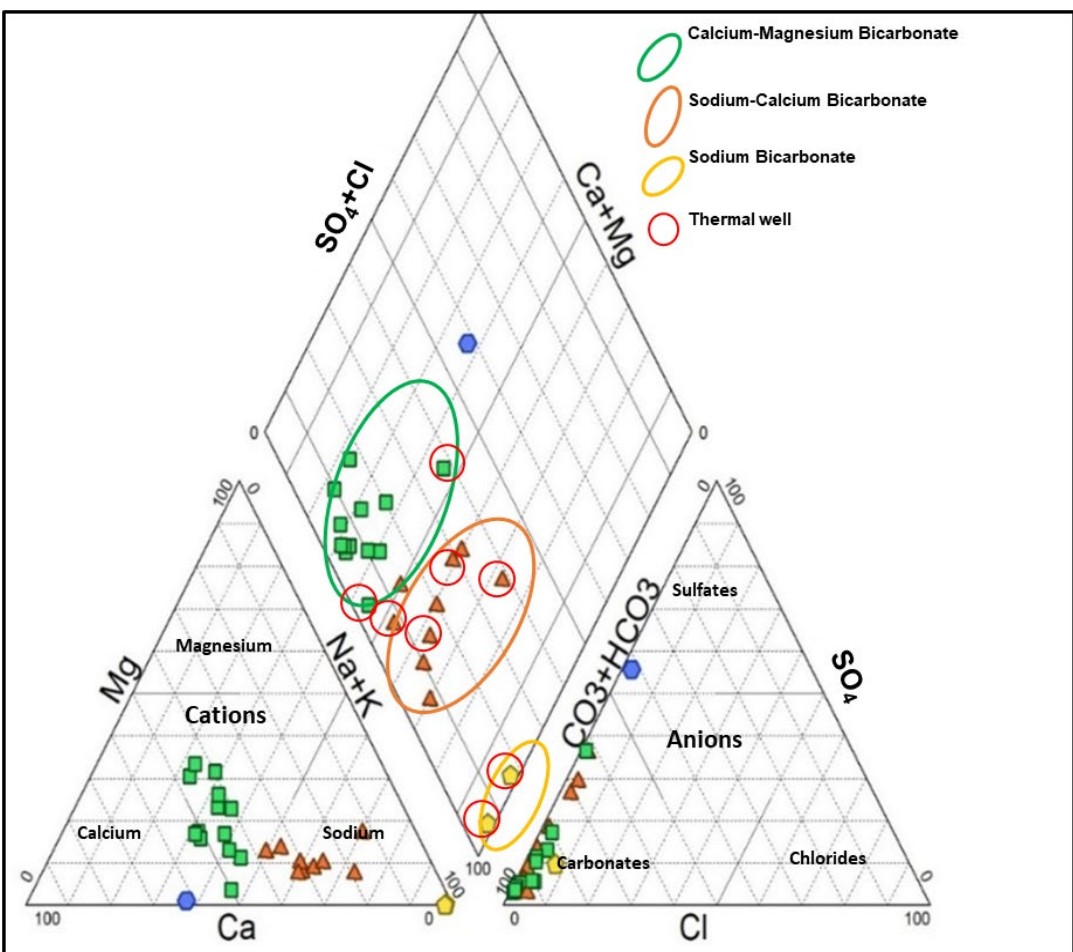

**Figure 7.** Piper diagram showing the hydrogeochemical facies or water families in wells located at the boundaries of the Silao-Romita, Valle de León, and La Muralla aquifers. The green boxes correspond to wells in the granular aquifers (Silao-Romita and Valle de León), the brown triangles correspond to wells in La Muralla aquifer and the yellow diamonds are a spring and a well with high thermalism. The blue hexagon indicates a well with a high sulfate concentration in the groundwater.

The thermal springs of the Comanjilla spa (M-22) and the well in El Salto de Abajo (M-14) have similar hydrogeochemical characteristics. This suggests that both localities are located on the trace of a fault, suggesting hydraulic connectivity via a regional or

intermediate flow via faults. The Comanjilla thermal spring and the El Salto de Abajo thermal well recorded high sodium (Na) contents, which indicates an ion exchange process, i.e., the water interacts with the rock constituents and therefore has a longer residence time.

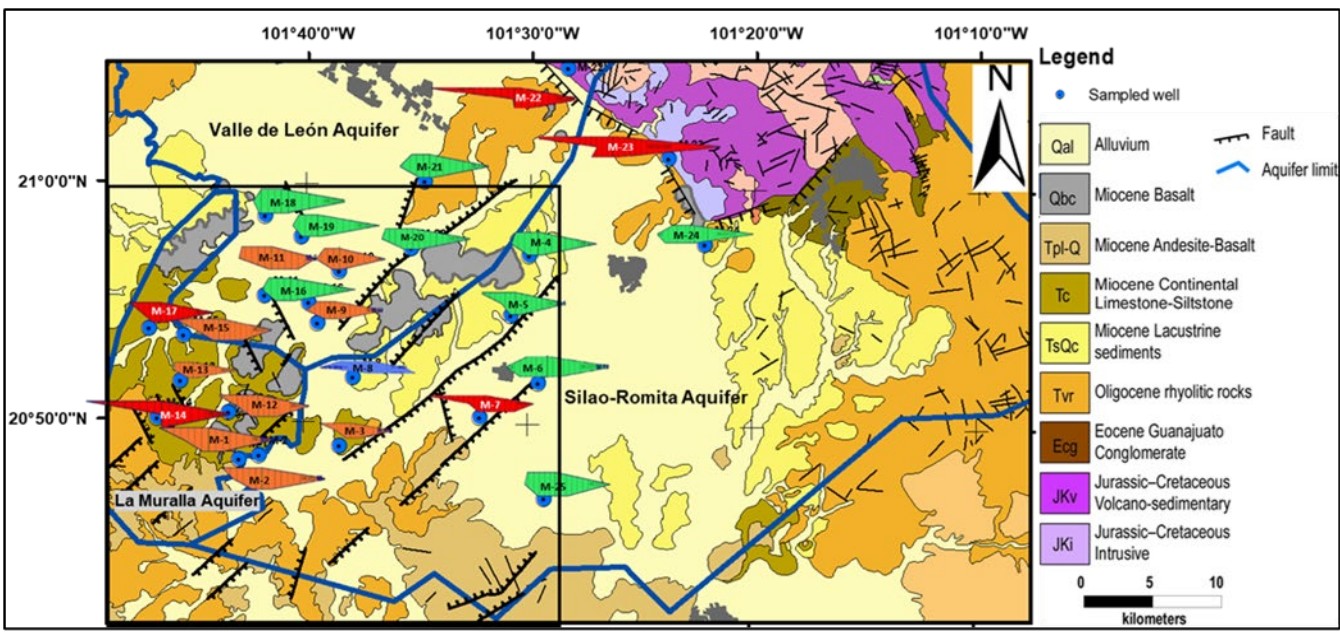

**Figure 8.** Distribution of Stiff plots representing the concentrations of major ions (anions and cations) of 24 wells and one spring in the study area. The figures in green are water of recent infiltration, those in orange correspond to water with low thermalism and in red correspond to wells with high thermalism located in fault zones.

According to the investigations in the surrounding areas, the groundwater with the highest temperature corresponds to bicarbonate facies (Na HCO$_3$), indicating that it has interacted with thermal fluids and could be controlled by faults and deep regional scale water flow [37]. The water temperature in these wells ranges between 36 and 50 °C [16]. It is also assumed that high-temperature wells are located near faults and fractures and coincide with high levels of other elements, such as arsenic and fluoride, as in this study region [15–17].

*3.4. Hydrogeochemical Processes*

The Gibbs diagram for the study area demonstrates that water–rock interaction is the primary process and that the weathering and erosion of the rocks control the groundwater chemistry.

The ratio of TDS to cations is depicted in Figure 9a. The ratio of TDS to anions is shown in Figure 9b, whereas the ratio of anions to chloride ions is low because of the high quantities of bicarbonate ions.

It was thought to be necessary to consider the connections between the principal ions and the regional geology to understand the hydrogeochemical processes better. Figure 10a illustrates a relationship between Na$^+$ and Cl$^-$. Although their ratio is not 1:1, as it would be in the case of halite dissolution, sodium is present in the groundwater in more significant concentrations than chloride, indicating that other mechanisms than halite dissolution are responsible. If the Na/Cl ratio is more significant than 1, it suggests silicate weathering, which is seen in most samples and is the cause of the high Na concentration. Extrusive igneous rocks, such as the nearby rhyolites, may have contributed to the high Na contents Figure 10b. A negative correlation is seen in the graph of Figure 10b, suggesting the presence of cation exchange mechanisms.

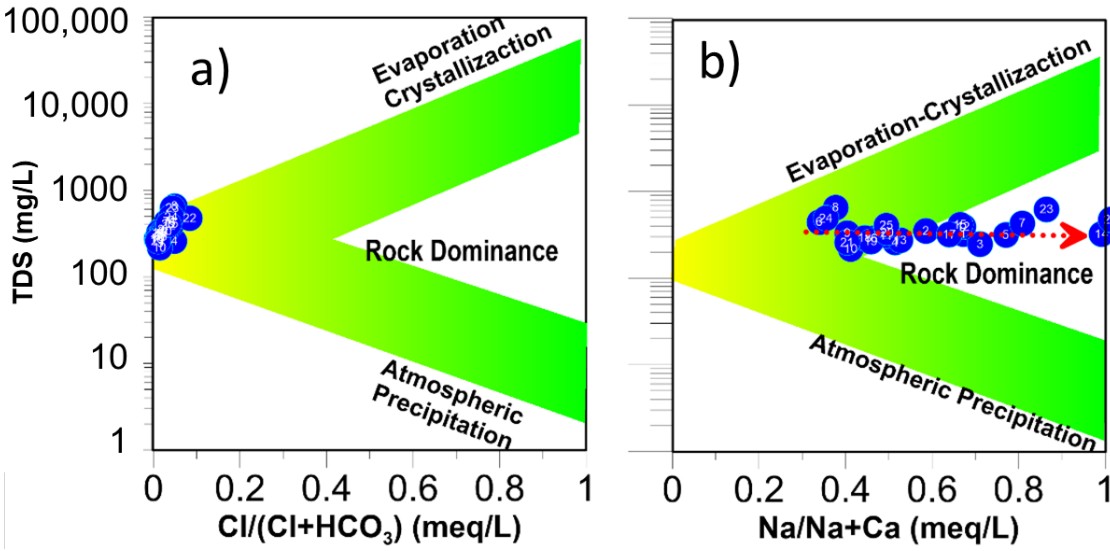

**Figure 9.** Gibbs diagram showing hydrogeochemical processes in groundwater. (**a**) Relationship between cations and Total Dissolved Solids (TDS). (**b**) Relationship between anions and Total Dissolved Solids (TDS). The arrow indicates the evolution of groundwater. The samples towards yellow are dominated by $HCO_3$ and Ca; while towards the green the Cl and Ca ions dominate. Basically, the water–rock interaction is observed.

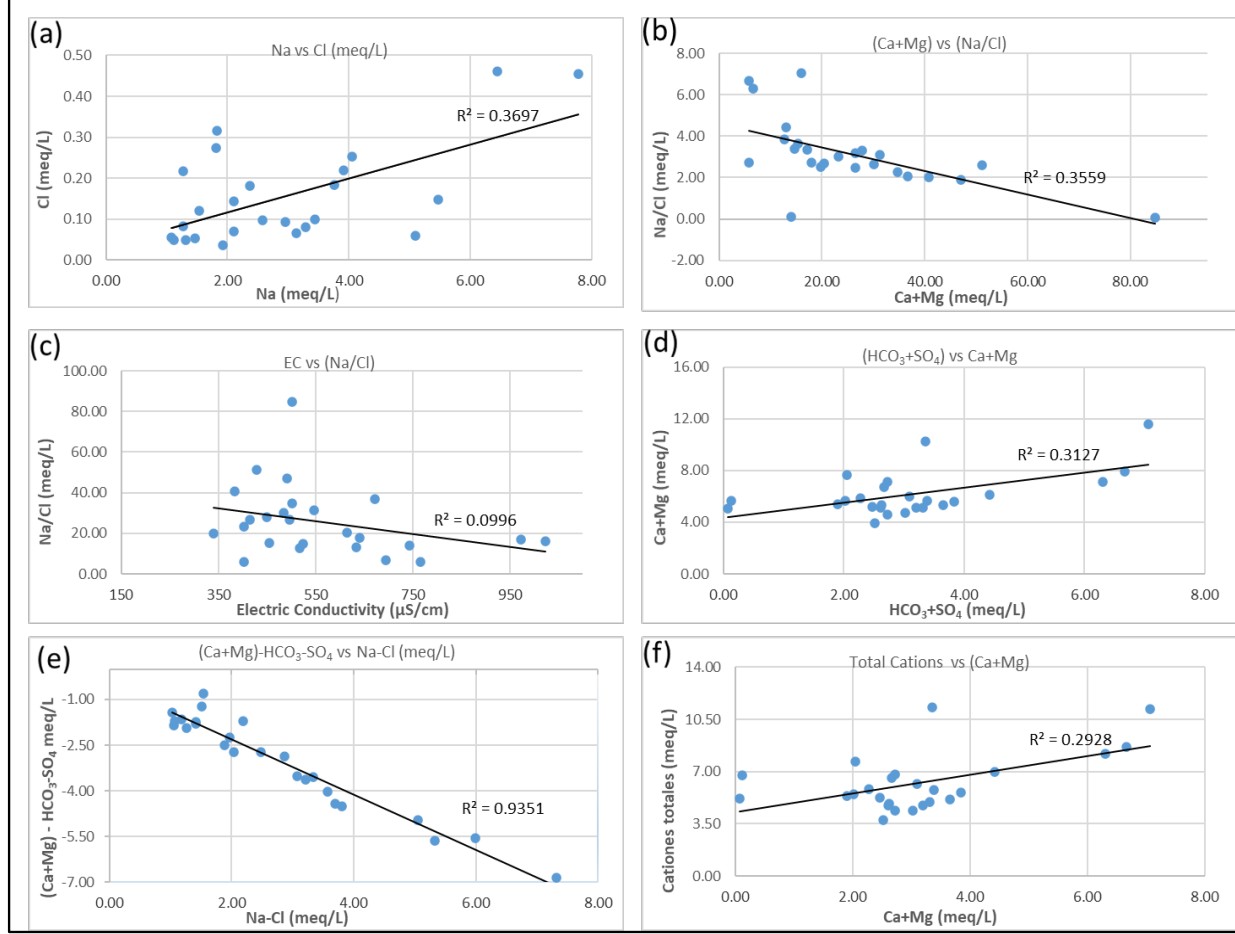

**Figure 10.** Relationships between (**a**) Na and Cl, (**b**) Ca + Mg and Na/Cl, (**c**) Electrical Conductivity and Na/Cl, (**d**) $HCO_3$ + $SO_4$ and Ca + Mg, (**e**) Na-Cl and (Ca + Mg)-($HCO_3$-$SO_4$), and (**f**) Ca + Mg and total cations.

### 3.5. Evaporation

Evaporation increases TDS, but the Na/Cl ratio is constant (Jankowski and Acworth, 1997). For the study area, the correlation is negative, the relationship between Na and Cl does not vary, and there is only an increase in the Electrical Conductivity (EC), as seen in the EC vs. Na/Cl graph (Figure 10c). Most of the samples in Figure 10d are below the mixing line, but 28% of them, or at least seven samples, are above it, showing normal and reverse ion exchanges. Normal is defined as having too much calcium and magnesium and coming from clay minerals with a sodium exchange. The relationship between Na-Cl and (Ca + Mg-$HCO_3$-$SO_4$) in Figure 10e is an example of a reverse exchange. When the slope of the trend line tends to be negative, a reverse ion exchange occurs in the groundwater. The inverse ion exchange mechanism confirms the link between Ca, Mg, and Na by $R^2 = 0.9351$.

### 3.6. Mineral Weathering

The weathering of silicate minerals is one of the primary sources of Na and K in groundwater. Pyroxenes, amphiboles, and calcic potassium feldspars are susceptible to weathering in common igneous rocks and can be a rich source of sodium and potassium [19]. Several clay compounds can be created from the silicate weathering result. The Figure 10a correlation plot between Na and Cl demonstrates that most points are in the Na zone instead of the Cl zone. This indicates that the study area has a high concentration of sodium ions. The correlation between Ca + Mg vs. total cations (Figure 10f) shows that silicate weathering has occurred.

Lacustrine materials are abundant in the study area, and they constitute clayey silt and calcareous materials, from which Ca and Mg can be added to the groundwater due to the dissolution of calcite. The ratio of Ca and Mg to $HCO_3$ was plotted to analyze calcite and dolomite dissolution (Figure 11a,b). The association of $HCO_3$ with magnesium is better than that with calcium. This means that calcium predominates over magnesium at several sites in the area, as seen in Table 1. This finding was confirmed by plotting Ca against Mg in Figure 11c, where most points are located below the 1:1 correlation line, indicating calcite dissolution. The points above the 1:1 correlation line indicate the dissolution of dolomite. If Ca/Mg = 1, this represents the dissolution of dolomite; if Ca/Mg is between 1 and 2, this represents the dissolution of calcite; and if it is greater than 2, which it is in this case, it represents silicate weathering [38]. A possible source of magnesium is the dissolution of aluminosilicates [39]. If Ca/Mg is less than 1, it indicates dolomite weathering; if it is between 1 and 2, it indicates calcite weathering; and if it is more significant than 2, in this instance, it means silicate weathering [38]. The dissolution of aluminosilicates has been suggested as a potential source of magnesium [39].

To identify potential sources of calcium and sulfate in the local groundwater, the Ca and $SO_4$ ions were plotted Figure 11d. Gypsum dissolving is a crucial source of Ca since there is a strong link between Ca and $SO_4$, which is observed. Additionally, it has been shown that sulfates exhibit a positive connection with sodium ions (Figure 11e), which implies that the mineralization of these elements, except for calcium, usually occurs. The research area may have buried evaporite sediments associated with lacustrine sediments as the source of this gypsum.

As a result of ion exchange processes between the porous medium and the aquifer waters, there is a decrease in calcium and magnesium in the K + Na-Cl vs. Ca + Mg-$SO_4$-$HCO_3$ ratio as a function of an increase in the sodium concentration (Figure 11f). Other authors have noted similar circumstances [19,38,39].

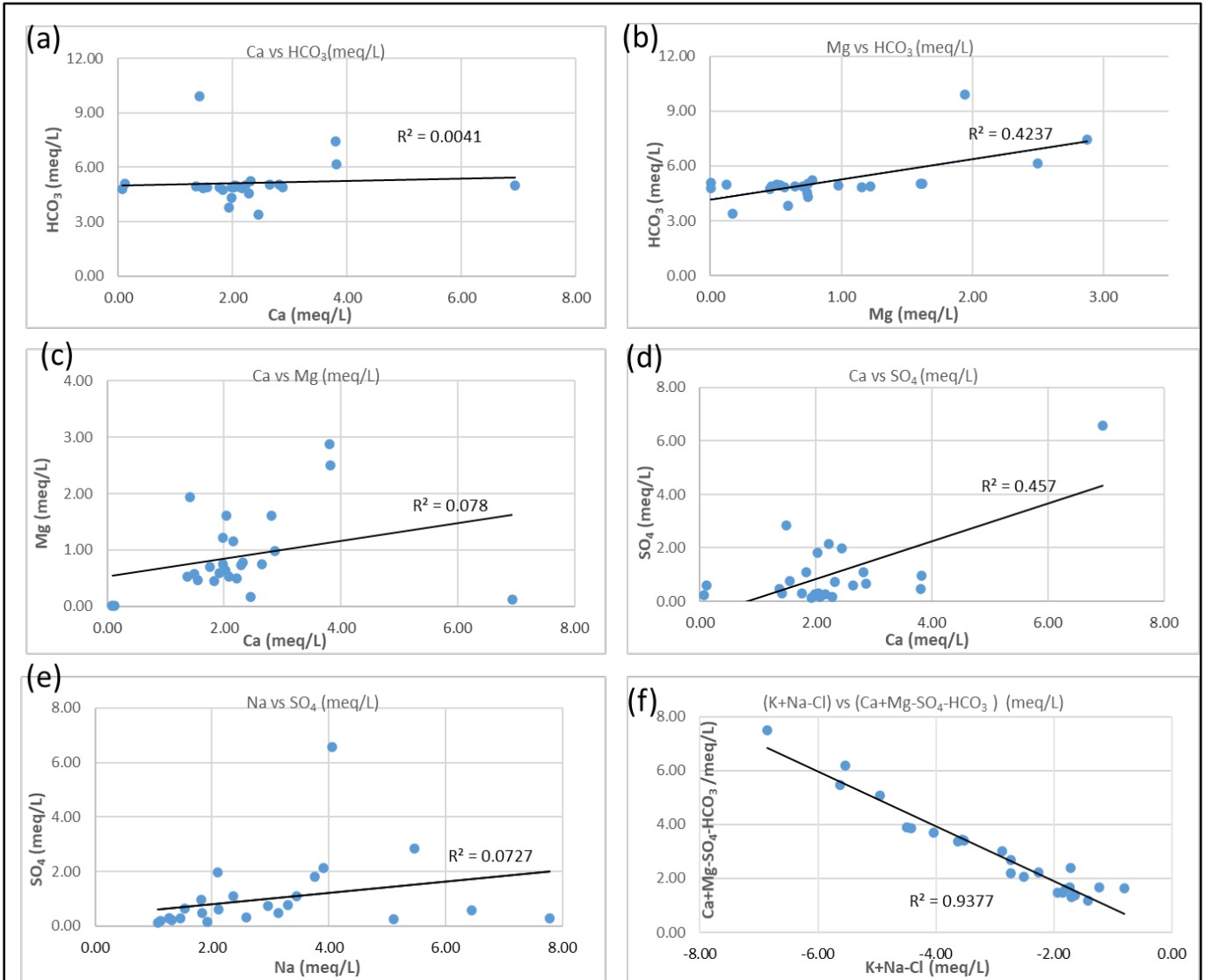

**Figure 11.** Relationship between (**a**) Ca and HCO₃, (**b**) Mg and HCO₃, (**c**) Ca and Mg, (**d**) Ca and SO₄, (**e**) Na and SO₄, (**f**) K + Na-Cl and K + Mg-SO₄-HCO₃.

### 3.7. Flow System Classification

Groundwater evolution can be shown in two tendencies in the classification of flow based on (Na + K) vs. (Cl + SO₄) [40] (Figure 12). The first is associated with a local flow that has experienced an increase in its primary ions, suggesting a potential connection to intermediate flows. The second trend is associated with an intermediate flow, where the temperatures of multiple wells rise, indicating a combination of thermal water at a deeper level associated with an intermediate flow and even associated with a regional flow. Regional faults that run from the northeast to southwest dominate some zones, suggesting that hydraulic connections between different aquifers may exist. Increases in some element concentrations may be related to regional and intermediate flows caused by fault networks and fractures with lengths of several kilometers [41].

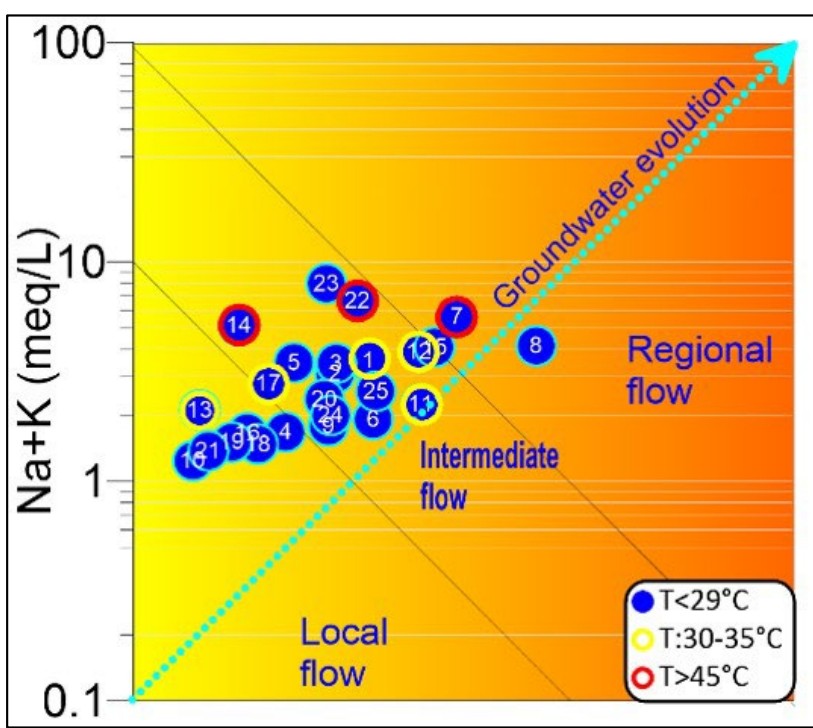

**Figure 12.** Diagram of Mifflin 1988, showing the type of groundwater flow based on the major ions. Blue color represents wells with T = 19.29 °C, yellow represents wells that are 30–39 °C, and red represents wells that are 41–95 °C.

## 3.8. Statistical Analysis

The Pearson's correlations for the pH, T °C, electric conductivity (EC), Total Dissolved Solids (TDS), Mg, Na. K, HCO$_3$, Cl, and SO$_4$ are shown in Table 3. The parameters that are significantly correlated at the 0.01 and 0.05 levels are written in bold. A very significant correlation can be noted between EC-TDS, HCO$_3$-Cl, and Mg-Na, whereas some of the couples, such as pH-Ca-Mg, have negative correlations. The highest correlation between EC-TDS with HCO$_3$, Cl, Mg, and Na suggests a water–rock interaction [37]. This corresponds to the geochemical explanations provided by the Gibbs diagrams (Figure 9). To examine the differences between the aquifers, we used variance tests, and we first used the Shapiro–Wilk hypothesis test with a 5% significance threshold to verify if our data set was normally distributed. Furthermore, because most parameters do not have a normal distribution, non-parametric tests were used to compare the median values and evaluate the three aquifers analyzed. A non-parametric test (Kruskal–Wallis) was used to evaluate whether there was a significant variation in the groundwater composition between the three aquifers examined (Table 4). The Kruskal–Wallis non-parametric test revealed substantial differences between the La Muralla and León aquifers. The most significant differences in order of significance ($p < 0.05$) are T °C, Na, and K; box plots are presented in the Supplementary Information. The La Muralla and Silao Romita aquifers, on the other hand, show more striking parallels in terms of significance (Table 4), with significant differences in terms of T °C ($p < 0.05$). The Cl levels in the Valle de Leon and Silao-Romita aquifers vary significantly ($p < 0.05$). This can also be explained by the aquifers' geological control and bounds. The geologies of Silao-Romita and La Muralla are comparable, with fractured rhyolites and continental deposits, as well as faults perpendicular to their boundaries. The Valle de Leon aquifer, on the other hand, differs considerably from the Silao-Romita and La Muralla aquifers because their boundaries correspond with faults subparallel to their limits, which can act as communication barriers between them. In summary, the variance analysis shows that the Silao-Romita and La Muralla aquifers are more connected, but the Valle de Leon aquifer has less communication with the others.

**Table 3.** Pearson's correlation coefficient matrix of physicochemical parameters in groundwater samples (*n* = 21). ** Correlation is significant at the 0.01 level (2-tailed). * Correlation is significant at the 0.05 level (2-tailed).

|  | pH | T °C | EC | TDS | Ca | Mg | Na | K | HCO$_3$ | Cl | SO$_4$ |
|---|---|---|---|---|---|---|---|---|---|---|---|
| **pH** | 1.00 | | | | | | | | | | |
| **T °C** | 0.08 | 1.00 | | | | | | | | | |
| **EC** | −0.38 | −0.15 | 1.00 | | | | | | | | |
| **TDS** | −0.38 | −0.15 | 1.00 ** | 1.00 | | | | | | | |
| **Ca** | −0.65 ** | −0.14 | 0.31 | 0.31 | 1.00 | | | | | | |
| **Mg** | −0.51 * | −0.47 * | 0.640 ** | 0.64 ** | 0.63 ** | 1.00 | | | | | |
| **Na** | 0.11 | 0.06 | 0.706 ** | 0.70 ** | −0.39 | 0.04 | 1.00 | | | | |
| **K** | 0.13 | −0.29 | −0.29 | −0.29 | −0.23 | −0.16 | −0.25 | 1.00 | | | |
| **HCO$_3$** | −0.36 | −0.37 | 0.86 ** | 0.86 ** | 0.12 | 0.69 ** | 0.68 ** | −0.16 | 1.00 | | |
| **Cl** | −0.33 | −0.25 | 0.90 ** | 0.90 ** | 0.30 | 0.64 ** | 0.61 ** | −0.08 | 0.83 ** | 1.00 | |
| **SO$_4$** | 0.01 | 0.49* | 0.25 | 0.25 | 0.17 | −0.24 | 0.27 | −0.42 | −0.19 | 0.15 | 1.00 |

**Table 4.** Results of the Kruskal–Wallis test (hierarchical one-way ANOVA) for the three studied aquifers. The comprehensive box plots from the multiple comparison Kruskal–Wallis test are available in Supplementary Information. The significance level is 0.05.

| Dependent Variable | Aquifer 1-Aquifer 2 | Test Statistic | Std. Error | Std. Test Statistic | Sig. | Adj. Sig. a |
|---|---|---|---|---|---|---|
| pH | La Muralla-Valle de León | −2.536 | 3.445 | −0.736 | 0.462 | 1.000 |
| | La Muralla-Silao-Romita | −3.688 | 3.344 | −1.103 | 0.270 | 0.811 |
| | Valle de León-Silao-Romita | 1.152 | 3.205 | 0.359 | 0.719 | 1.000 |
| T °C | Valle de León-Silao-Romita | 3.759 | 3.206 | 1.172 | 0.241 | 0.723 |
| | Valle de León-La Muralla | 10.988 | 3.446 | 3.188 | 0.001 | 0.004 |
| | Silao-Romita-La Muralla | 7.229 | 3.346 | 2.161 | 0.031 | 0.092 |
| EC | Valle de León-Silao-Romita | 5.330 | 3.210 | 1.660 | 0.097 | 0.290 |
| | Valle de León-La Muralla | 5.643 | 3.451 | 1.635 | 0.102 | 0.306 |
| | Silao-Romita-La Muralla | 0.313 | 3.350 | 0.093 | 0.926 | 1.000 |
| TDS | Valle de León-Silao-Romita | 5.330 | 3.210 | 1.660 | 0.097 | 0.290 |
| | Valle de León-La Muralla | 5.643 | 3.451 | 1.635 | 0.102 | 0.306 |
| | Silao-Romita-La Muralla | 0.313 | 3.350 | 0.093 | 0.926 | 1.000 |
| Ca | La Muralla-Silao-Romita | −1.625 | 3.351 | −0.485 | 0.628 | 1.000 |
| | La Muralla-Valle de León | −2.643 | 3.452 | −0.766 | 0.444 | 1.000 |
| | Silao-Romita-Valle de León | −1.018 | 3.211 | −0.317 | 0.751 | 1.000 |
| Mg | La Muralla-Silao-Romita | −5.417 | 3.351 | −1.616 | 0.106 | 0.318 |
| | La Muralla-Valle de León | −6.310 | 3.452 | −1.828 | 0.068 | 0.203 |
| | Silao-Romita-Valle de León | −0.893 | 3.211 | −0.278 | 0.781 | 1.000 |
| Na | Valle de León-Silao-Romita | 7.232 | 3.211 | 2.252 | 0.024 | 0.073 |
| | Valle de León-La Muralla | 10.857 | 3.452 | 3.145 | 0.002 | 0.005 |
| | Silao-Romita-La Muralla | 3.625 | 3.351 | 1.082 | 0.279 | 0.838 |
| K | La Muralla-Silao-Romita | −3.917 | 3.351 | −1.169 | 0.242 | 0.727 |
| | La Muralla-Valle de León | −8.024 | 3.452 | −2.324 | 0.020 | 0.060 |
| | Silao-Romita-Valle de León | −4.107 | 3.211 | −1.279 | 0.201 | 0.603 |
| HCO$_3$ | Valle de León-La Muralla | 1.702 | 3.445 | 0.494 | 0.621 | 1.000 |
| | Valle de León-Silao-Romita | 3.411 | 3.205 | 1.064 | 0.287 | 0.862 |
| | La Muralla-Silao-Romita | −1.708 | 3.344 | −0.511 | 0.609 | 1.000 |
| Cl | Valle de León-La Muralla | 4.190 | 3.452 | 1.214 | 0.225 | 0.674 |
| | Valle de León-Silao-Romita | 6.982 | 3.211 | 2.174 | 0.030 | 0.089 |
| | La Muralla-Silao-Romita | −2.792 | 3.351 | −0.833 | 0.405 | 1.000 |
| SO$_4$ | Valle de León-Silao-Romita | 6.304 | 3.211 | 1.963 | 0.050 | 0.149 |
| | Valle de León-La Muralla | 7.095 | 3.452 | 2.055 | 0.040 | 0.120 |
| | Silao-Romita-La Muralla | 0.792 | 3.351 | 0.236 | 0.813 | 1.000 |

### 3.9. Deuterium and Oxygen Isotopic Composition

The isotopic results of 17 samples are shown in Table 5. The spatial distribution and their positions concerning the Global Meteoric Line (GMWL) and the Local Meteoric Line (LMWL) are shown in Figure 13a. Some samples are located on the LMW; others are associated with evaporation processes related to surface water bodies and evaporation processes due to irrigation returns. The temperature in the thermal wells also modifies the values of the isotopic signature in the zone.

**Table 5.** Isotopic results of deuterium ($\delta\,^2$H) and oxygen 18 ($\delta\,^{18}$O) in the groundwater of the study area.

| ID | Coordinates | | Physico-Chemical Parameters | | | | Isotopes | |
|---|---|---|---|---|---|---|---|---|
| | Latitude | Longitude | pH | T (°C) | EC (µS/cm) | TDS (mg/L) | $\delta^{18}$O (‰) | $\delta^2$H (‰) |
| M-1 | 20°48′24″ | 101°42′47″ | 7.55 | 30.5 | 502 | 321.28 | −76.520915 | −10.284116 |
| M-3 | 20°49′01″ | 101°38′20″ | 7.61 | 28.7 | 384 | 245.76 | −76.8683705 | −10.320088 |
| M-5 | 20°54′36″ | 101°30′46″ | 7.61 | 25.8 | 491 | 314.24 | −82.3333356 | −10.992 |
| M-7 | 20°50′17″ | 101°32′07″ | 7.51 | 30.9 | 671 | 429.44 | −71.430466 | −9.4682802 |
| M-8 | 20°51′54″ | 101°37′46″ | 7.28 | 25 | 1023 | 654.72 | −74.995188 | −9.6851966 |
| M-10 | 20°56′21″ | 101°38′27″ | 7.7 | 24.5 | 340 | 217.6 | −83.4117049 | −11.213424 |
| M-11 | 20°55′17″ | 101°41′46″ | 7.58 | 38.6 | 484 | 309.76 | −75.4597666 | −10.412072 |
| M-12 | 20°50′ 22″ | 101°43′18″ | 7.49 | 31.4 | 614 | 392.96 | −75.4023192 | −9.8022843 |
| M-13 | 20°51′38″ | 101°45′27″ | 7.28 | 30.4 | 428 | 273.92 | −79.1533606 | −9.978976 |
| M-14 | 20°50′04″ | 101°46′27″ | 9.03 | 41.3 | 502 | 321.28 | −79.6393628 | −10.510739 |
| M-17 | 20°53′50″ | 101°46′51″ | 7.49 | 35.9 | 497 | 318.08 | −77.339109 | −10.048801 |
| M-18 | 20°58′38″ | 101°41′48″ | 7.47 | 26.2 | 455 | 291.2 | −83.208591 | −10.804766 |
| M-20 | 20°57′21″ | 101°35′15″ | 7.36 | 25.3 | 524 | 335.36 | −75.3109938 | −9.4467074 |
| M-22 | 21°05′01″ | 101°28′22″ | 8.71 | 95 | 743 | 475.52 | −77.1162215 | −9.8375429 |
| M-23 | 21°01′16″ | 101°23′48″ | 7.53 | 21.8 | 973 | 622.72 | −77.5618975 | −10.059836 |
| M-24 | 20°57′41″ | 101°22′09″ | 7.17 | 26.2 | 765 | 489.6 | −78.7530691 | −10.071817 |
| M-25 | 20°46′54″ | 101°29′10″ | 7.56 | 25.9 | 633 | 405.12 | −76.1983787 | −10.093386 |

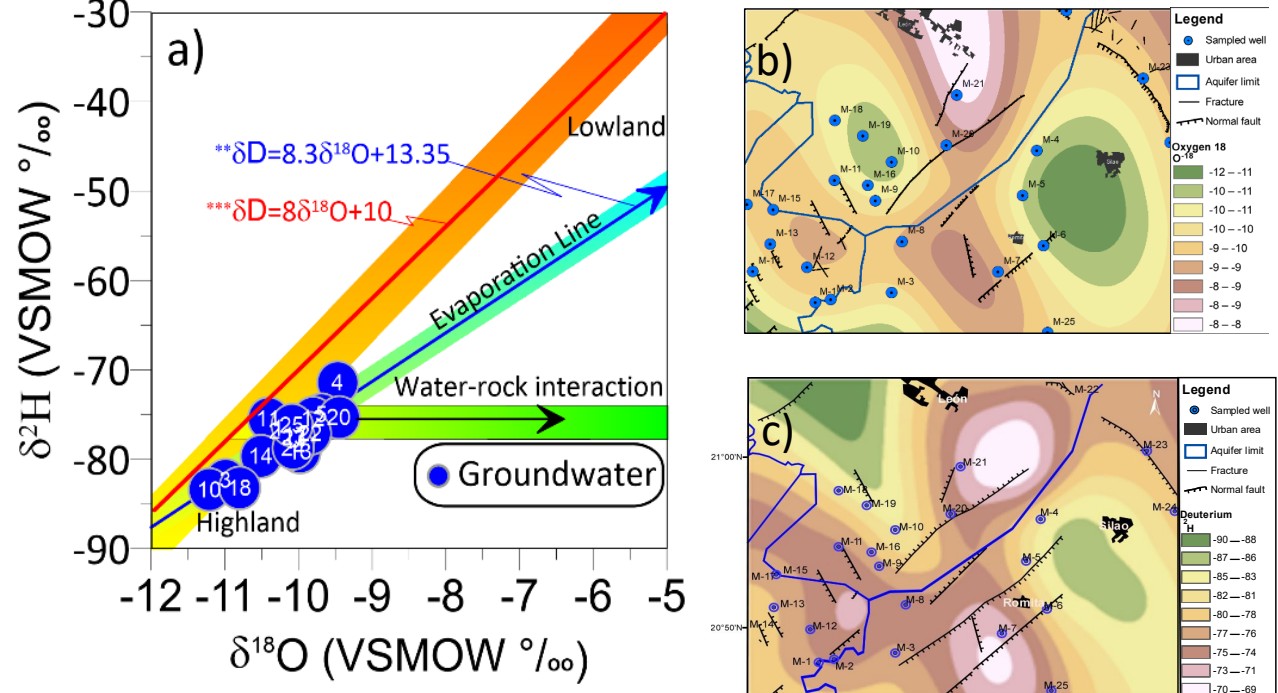

**Figure 13.** (**a**) Graph showing the stable isotope ratio of oxygen-18 and deuterium, (**b**) the spatial distribution of oxygen-18, (**c**) the spatial distribution of deuterium in the study area. ** In blue groundwater evaporation line. *** In red Global Meteoric Line (GMWL).

The isotopic results of $\delta^{18}O$ and deuterium ($\delta^2H$), with ranges of $-83.41$ to $-71.43$ and $-11.21$ to $-9.45$, indicate that part of the groundwater volume has been exposed to evaporation processes due to the presence of surface water bodies and irrigation returns. The zones with the highest isotopic enrichment are in recharge zones. In contrast, the most impoverished zones are situated in the valleys, with a more significant water–rock interaction and a longer residence time, suggesting that the local flow waters are mixed with deeper or intermediate flows [42] (Figure 13b,c).

## 4. Conclusions

Three hydrogeochemical facies were identified (calcium–magnesium bicarbonate, sodium–calcium bicarbonate, and sodium bicarbonate) at the boundaries of the Silao-Romita, the Valle de León, and the La Muralla aquifers. The hydrogeochemical characterization and processes imply hydraulic linkage via regional thermal flows enhanced by faults and the mixing of local flow waters with intermediate flows. This is evidenced by the similar hydrochemical signatures of the thermal spring and deep thermal well. The isotopic results indicate that part of the groundwater volume has been exposed to local evaporation processes due to the presence of surface water bodies and irrigation returns. The highest isotopic enrichment is observed near or in the recharge regions. In contrast, the most depleted zones are in the valleys, where there is more a significant interaction with the rock and a longer residence time, implying a mixture of the local water flows with deeper or intermediate flows, which, when combined with water geochemistry, indicates a connection between the aquifers studied. The results of the Kruskal–Wallis non-parametric variance test reveal substantial differences between the La Muralla and Valle de León aquifers. The most significant differences ($p < 0.05$) are in the T $°C$, Na, and K values. This can also be explained by the aquifers' geological control and bounds. In summary, the variance analysis shows that the Silao-Romita and La Muralla aquifers are more connected, but the Valle de Leon aquifer communicates less with the others.

In conclusion, we confirm the hypothesis by identifying the hydrogeochemical processes, the relationship, and the mixing between the groundwater of the fractured aquifer (La Muralla) and the granular aquifers (Valle de León and Silao-Romita).

The information generated in this study is of great value for one of the most populated and economically active areas in central Mexico.

**Supplementary Materials:** The following supporting information can be downloaded at: https://www.mdpi.com/article/10.3390/w15223948/s1.

**Author Contributions:** Conceptualization, G.I.-O., R.M.-A., J.A.R.-L. and M.J.P.-A.; methodology, G.I.-O., R.M.-A. and J.M.-R.; software, Y.L. and P.K.; formal analysis, G.I.-O. and R.M.-A.; investigation, R.M.-A., G.I.-O. and J.A.R.-L.; writing—original draft preparation, G.I.-O., R.M.-A. and P.K.; writing—review and editing, G.I.-O., R.M.-A and E.Á.-M. All authors have read and agreed to the published version of the manuscript.

**Funding:** The Mexican Geological Survey, University of Guanajuato project CIIC 2023-031/2023, and CONACyT funded this study.

**Data Availability Statement:** The data produced by this study are available upon request to the authors.

**Acknowledgments:** The authors thank CEAG for their supporting information.

**Conflicts of Interest:** The authors declare no conflict of interest.

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
