# Peer review of "Hydrogeochemical Characterization of Groundwater at the Boundaries of Three Aquifers in Central México"

_water, doi:10.3390/w15223948_

Round 1
Reviewer 1 Report
Comments and Suggestions for Authors
This manuscript is not ready for publication in the journal Water. In the methodological chapter, the description of some of the methods used in the work is missing or incomplete. Furthermore, I have some presentation concerns.
General comments:
One of the main goals of this work is to connect the hydrogeochemical composition of groundwater with local geological and hydrogeological features. The article lacks a geological map, a hydrogeological map and a hydrogeological profiles (at least two) through the observed area. Indications of the geological map are visible in Figure 6, however, the description of individual geological units shown in the legend (on the figure or in the text) is missing. Figures 2, 3 and 4 show the spatial distribution of individual ions in groundwater, without explaining which interpolation method was used to obtain these maps. This needs to be clarified in the methodological part.
In the results and discussion chapter, a broader and more precise interpretation of the results of all conducted analyses, in relation to local geological and hydrogeological characteristics, is missing. The impression is that the authors gave interpretations of individual, partial analyses, but did not make a synthesis of all results through certain conclusions that would be compared with research in areas with similar geological and hydrogeological research. Literature sources are limited, some are incomplete and are not presented in accordance with the instructions for authors (for example 2, 4, 21, 31, 36).
Specific comments:
· All maps lack the scale and north markings
· The text in the legend in Figure 1 is poorly visible, it is necessary to increase the font
· Line 95: „The 95 Sierra de Guanajuato has the most insufficient amounts of actual evapotranspiration“. Please explain the statement: “insufficient amount of actual evapotranspiration”! What does it mean in the context of your work?
· Line 144: “…to observe the distribution of wells and the relationship between aquifers, (Figure 2)“. In Figure 2, not all wells are shown, nor is the relationship between individual aquifers defined!
Line 147: „..In Figure 2, not all wells are shown, nor is the relationship between individual aquifers defined.“. Figure 3 does not clearly show the monitoring program that is carried out at the borders of the three aquifers. It is necessary to clearly mark and distinguish individual aquifers.
· Line 153; Line 155: “The monitoring networks of the three aquifers comprise 64 wells, but only 14 are in interest… A total of 24 deep wells and one thermal spring were sampled for physicochemical analysis“. If 14 wells are interesting (why not defined), why were 24 wells sampled? What were the criteria for determining the importance of individual wells? It is necessary to explain further. It is not clear for which period the water samples were analyzed. It is common practice to repeat the sampling at least two to four times during the hydrological year when determining the hydrogeochemical characteristics of groundwater, in order to determine the natural variability of groundwater composition over time. It is not documented here nor is it explained why it was not implemented.
· Line 176:“ Bond and their respective isotopic combinations since the laser is adjusted to the frequency of the H-O bond of the water vapor molecules (patented technology).“ This sentence is not understandable and needs to be written differently. If it is a patented technology, it must be described and documented in a few sentences.
· Line 185: „The Electrical Conductivity (EC) values vary between 340 and 1023 µS/cm, with an average of 564 µS/cm, indicating good water quality.“ Why do these indicators indicate good water quality? They indicate a certain degree of mineralization. If water of a certain quality for a very specific purpose was meant here, then it should be specified.
· In the legends on figures 2, 3, and 4, the measurement units of individual ions in water are missing.
· Some labels are missing in the legend of Figure 5.
· Line 212: “due to water-rock interaction and the direction of subsurface flow from north to south. The spatial variation of the STD diagram“…What is STD diagram??
· Line 322: “6 samples correspond to thermal wells with temperatures of 30 to 35 °C, which shows a process of cation exchange and water-rock interaction“ What are those wells? They must be clearly marked on the diagram.
· Line 329: „According to the Stiff diagrams and the geology of the study area (Figure 6)…“. The Stiff diagrams in Figure 6 are unclear and nothing can be inferred from the figure. It is necessary to find some other graphical solution so that Stiff's diagrams are clearly visible.
· Line 335: “The thermal springs of the Comanjilla spa (M-22) and the well in El Salto de Abajo (M-14) have similar hydrogeochemical characteristics“. Where are those wells? It is necessary to clearly mark them on figures, especially on figures 5 and 6.
· Line 342: “According to investigations in the surrounding areas, the groundwater with the highest temperature correlates to the sodium bicarbonate facies (Na HCO3), indicating that it has interacted with thermal fluids.“ It is written too generally and it needs to be further documented in the text. It is necessary to mark the wells with high temperature separately in Figures 5 and 6.
· Line 392: “For the study area, the correlation is negative, the relationship between Na and Cl does not vary,…“ It is evident from the figure that the correlation is weak and such conclusions cannot be drawn. Furthermore, there is still a certain variability of Na and Cl…
· The mark of the vertical axis in Figure 8e is barely visible. Why is the coefficient of determination shown in this figure, but not in the others?
· Line 411: “Figure 8(f) correlation plot between Na and Cl demonstrates that most points are in the Na zone instead of the Cl zone“. Figure 8 f shows something completely different!
· On page 12, an error appears: (Error! Reference source not found!)
· Figure 11 a is of very poor resolution and quality. It is necessary to make a changes!!
· Figures 11 b and c are too small, nothing can be seen. Certain structural-geological elements that appear on the map are not shown in the legend. North markings and graphic scale are missing.
Comments on the Quality of English Language
Some sentences should be written more clearly (marked in the reviewer's remarks).
Author Response
Dear Editor and reviewer, I trust this letter finds you well. I wish to extend my sincere appreciation for the time and expertise you invested in evaluating our manuscript titled " Hydrogeochemical Characterization of Groundwater at the Boundaries of Three Aquifers in Central México " submitted to the journal Water, Special Issue "Application of Integrated Geophysical, Hydrogeological and Geospatial Approach to Groundwater Exploration and Contamination". Your thorough assessment and insightful comments have been immensely valuable in refining our work and ensuring its scholarly rigor. Furthermore, your recognition and encouragement inspire us to improve our manuscript even further. We have carefully considered each of your suggestions and critiques, and I am pleased to provide our detailed responses below: Response to Reviewer 1 1. General comments for Authors: One of the main goals of this work is to connect the hydrogeochemical composition of groundwater with local geological and hydrogeological features. The article lacks a geological map, a hydrogeological map and a hydrogeological profiles (at least two) through the observad area. lndications of the geological map are visible in Figure 6, however, the description of individual geological units shown in the legend (on the figure or in the text) is missing. Figures 2, 3 and 4 show the spatial distribution of individual ions in groundwater, without explaining which interpolation method was used to obtain these maps. This needs to be clarified in the methodological part. In the results and discussion chapter, a broader and more precise interpretation of the results of all conducted analyses, in relation to local geological and hydrogeological characteristics, is missing. The impression is that the authors gave interpretations of individual, partial analyses, but did not make a synthesis of all results through certain conclusions that would be compared with research in areas with similar geological and hydrogeological research. Literatura sources ara limitad, sorne are incompleta and are not presentad in accordance with the instructions for authors (for example 2, 4, 21, 31, 36). General statements: Dear reviewer, we sincerely appreciate your thorough review and insightful feedback on our manuscript. Your thoughtful assessment provides valuable perspectives for enhancing the depth and impact of our work. We had the geologic map with sections; we put the geologic units in the legend and updated the methodology, including the method of interpolation employed, thanks to your comments. Statistical analyses and additional literature supported the discussion and results. We will respond individually to your comments and suggestions in the following section.
|
2. Point-by-point response to Comments and Suggestions for Authors
|
Comment 1: All maps lack the scale and north markings. |
Response 1: Thank you for your observation. The north and the graphic scale have been added to all the corresponding maps (see figures 1, 2, 3, pages 3,4 and 6).
|
Comment 2: The text in the legend in Figure 1 is poorly visible It is necessary to increase the font. |
Response 2: The font was increased, and the graphic scale and north were added (See Figure 1).
Comment 3: Line 95: “The Sierra de Guanajuato has the most insufficient amounts of actual evapotranspiration. Please explain the statement: “Insufficient amounts of actual evapotranspiration”! What does it mean in the context of your work? Response 3: Thank you for your observation. We corrected the wording and meaning of the sentence used. The idea is to highlight that due to climatic and geological conditions, the Sierra de Guanajuato is an important recharge zone for the Valle de León and Silao-Romita aquifers.
Comment 4: Line 144: „…to observe the distribution of wells and the relationship between aquifers,(Figure 2)”. In Figure 2, not all wells are shown nor is the relationship between individual aquifers defined! Response 4. Thank you for your observation. The total distribution of wells is shown in Figure 1, not Figure 2. In the text, the word "relationship" was replaced by "boundaries" since the figure shows the distribution of wells and boundaries between aquifers, not their relationship (see line 164).
Comment 5: Line 147: „…. In Figure 2, not all wells are shown, nor is the relationship between individual aquifers defined”. Figure 3 does not clearly show the monitoring program that is carried out at the borders of the three aquifers. It is necessary to clearly mark and distinguish individual aquifers. Response 5. Figure 2 was added to show the distribution of 24 wells and 1 thermal spring selected for sampling (Sampling Program).
Comment 6: Line 153; Line 155: “The monitoring networks of the three aquifers comprise 64 wells, but only 14 are in interest… A total of 24 deep wells and one thermal spring were sampled for physicochemical analysis”. If 14 wells are interesting (why not defined), why were 24 wells sampled?, What were the criteria for determining the importance of individual wells? It is necessary to explain further. It is not clear for which period the water samples were analyzed. It is common practice to repeat the sampling at least two to four times during the hydrological year when determining the hydrogeochemical characteristics of groundwater, in order to determine the natural variability of groundwater composition over time. It is not documented here nor is it explained why it was not implemented. Response 6: The basic criteria for selecting the sampling sites were: a) wells located at the boundaries of the granular aquifers and the fractured aquifer, b) deep wells with a piezometric level representative of the study area, c) preferably with a piezometric history, d) wells located in fault and thermal zones, and e) wells located in recharge zones. The selected period was rainfall 2022, considering the analysis of CEAG information from at least 3 previous years of the same period (see lines 160-165, page 5).
Comment 7: Line 176: “Bond and their respective isotopic combinations since the laser is adjusted to the frequency of the H-O bond of the water vapor molecules (patented technology)”. This sentence is not understandable and needs to be written differently. If it is a patented technology, it must be described and documented in a few sentences. Response 7: The information was supplemented and the sentence was modified.
Comment 8: Line 185: „ The electrical Conductivity (EC) Values vary between 340 and 1023 µS/cm, with an average of 564 µS/cm, indicating good water quality.” Why do these indicators indicate good water quality? They indicate a certain degree of mineralization. If water of a certain quality for a very specific purpose was meant here, then it should be specified Response 8: Totally agree. This is a parameter (EC) related to Total Dissolved Solids, which are low in this study (250 to 500 mg/L). The Mexican Official Standard indicates a maximum of 1000 (mg/L). However, this is correct, a single parameter does not indicate good water quality. This sentence was deleted in the text line 215.
Comment 9: In the legend on figures 2,3 and 4, the measurement units of individual ions in water are missing Response 9: The units of the ions in mg/L were added to the (see Figures 2, 3 and 4).
Comment 10: Some labels are missing in the legend of figure 5. Response10: Labels were added (see Figure 5) for the water families.
Comment 11: Line 212: “due to water-rock interaction and the direction of subsurface flow from north to south. The spatial variation of the STD diagram”…. What is STD diagram? Response11: We appreciate your observations. The diagram corresponds to Figure 2 (TDS and not STD). In the text STD was replaced by TDS. See lines, 342, page 13
Comment 12: Line 322: “6 samples correspond to thermal wells with temperatures of 30 to 35 °C, which shows a process of cation exchange and water-rock interaction” What are those wells? They must be clearly marked on the diagram. Response12: The 6 thermal wells were marked in the figure and legend8(see figure 7, page 12 and figure 12, page 17).
Comment 13: Line 329: According to the Stiff diagrams and the geology of the study area (Figure 6)…” The Stiff diagrams in Figure 6 are unclear and nothing can be inferred from the figure. It is necessary to find some other graphical solution so that Stiffs diagrams are clearly visible. Response13: Thank you for your observation Figure 8 was modified in order to clearly observe the Stiff diagrams and the mixing zone between water families or hydrogeochemical facies.
Comment 14: Line 335: “The thermal spring of the Comanjilla spa (M-22) and the well in El Salto de Abajo (M-14) have similar hydrogeochemical characteristics”. Where are those wells? It is necessary to clearly mark them on figures, especially on figures 5 and 6. Response14: According to your suggestion, both wells are marked in Figures 8,page 12 and 12 page 17. It is worth mentioning that for Figure 5, both the Comanjilla spring and the El Salto de Abajo well are the only two wells that belong to the sodium bicarbonate family and that present high thermalism.
Comment 15: Line 342. “According to investigations in the surrounding areas, the groundwater with the highest temperature correlates to the sodium bicarbonate facies (Na- HCO3, indicating that it has interacted with thermal fluids.” It is written too generally and it needs to be further documented in the text. It is necessary to mark the wells with high temperature separately in figures 5 and 6. Response 15: The thermal spring and the well with the highest thermalism were marked (see Figures 8 and 12).
Comment 16: Line 392. “For the study area, the correlation is negative, the relationship between Na and Cl does not vary,… It is evident from the figure that the correlation is weak and such conclusions cannot be drawn. Furthermore, there is still a certain variability of Na and Cl… Response16: This sentence was eliminated “This means that evaporation is not a relevant factor in the ground water chemistry”.
Comment 17: The mark of the vertical axis in Figure 8(e) is barely visible. Why is the coefficient of determination shown in this figure, but not in the others? Response17: Thank you for your observation. This diagram shows the coefficient of determination for the relationship between Ca+Mg-HCO3--SO4 as a function of Na-Cl-, which is useful to visualize if there is cation exchange in aquifers. For this reason, the coefficient of determination was calculated in this diagram (the R2 is shown in all the graphs).
Comment 18: Line 411. “Figure 8(f) correlation plot between Na and Cl demonstrates that most points are in the Na zone instead of the Cl zone”. Figure 8(f) shows something completely different!. Response18: Thank you for your observation. That is correct, the letter (f) was changed to (a).
Comment 19: On page 12, an error appears: (Error! Reference source not found!) Response19: Errors corrected. Figure linking errors.
Comment 20: Figure 11 (a) is of very poor resolution and quality. It is necessary to make a changes Response20: Figure 11(a) was produced with higher resolution.
Comment 21: Figures 11 (b) y (c) are too small, nothing can be seen. Certain structural-geological elements that appear on the map are not shown in the legend. North markigs and graphic scale are missing. Response21: Thank you for your comments. The maps have been edited again with a larger size. North and graphic scale were included in all maps.
|
4. Response to Comments on the Quality of English Language |
Point 1: |
Response 1: An English speaker revised the English to improve the manuscript. |
Once again, thanks a lot for your review. Your guidance will undoubtedly contribute to improving the overall quality of our manuscript.

Reviewer 2 Report
Comments and Suggestions for Authors
In this study, the authors investigated “Hydrogeochemical Characterization of Groundwater at the Boundaries of Three Aquifers in Central México”. Firstly, the authors identified three Three hydrogeochemical parameters, namely, calcium-magnesium bicarbonate, sodium-calcium bicarbonate, and sodium bicarbonate. What is the difference of three study areas? one-way analysis of variance (ANOVA) is used in the manuscript. Secondly, there are types of this parameters were the “Groundwater Types, Hydrogeochemical Processes, Evaporation, low System Classification Deuterium and Oxygen Isotopic Composition” sectors. Are they including the hydrogeochemical characterization? Thirdly, in the”Material and methods” section, where are the statistical analysis ? Thus, I suggest major revisions, and then accepted it.
Comments on the Quality of English LanguageMinor editing of English language required
Author Response
Dear Editor and reviewer, I trust this letter finds you well. I wish to extend my sincere appreciation for the time and expertise you invested in evaluating our manuscript titled " Hydrogeochemical Characterization of Groundwater at the Boundaries of Three Aquifers in Central México " submitted to the journal Water, Special Issue "Application of Integrated Geophysical, Hydrogeological and Geospatial Approach to Groundwater Exploration and Contamination". Your thorough assessment and insightful comments have been immensely valuable in refining our work and ensuring its scholarly rigor. Furthermore, your recognition and encouragement inspire us to improve our manuscript even further. We have carefully considered each of your suggestions and critiques, and I am pleased to provide our detailed responses below:
3. Point-by-point response to Comments and Suggestions for Authors
|
Comment 1: In the study investigated “Hydrogeochemical Characterization of Groundwater at the Boundaries of Three Aquifers in Central México”. Firstly, the authors identified three, Three hydrochemical parameters, namely, calcium-magnesium bicarbonate, sodium-calcium bicarbonate, and sodium bicarbonate. What is the difference of three study areas? One-way analysis of variance (ANOVA) is used in the manuscript. Response 1: Thank you very much for your suggestions. We took it into account, initially performing a normality test on the data, finding that most of the parameters do not have a normal distribution, non-parametric tests were used to compare the mean values and evaluate the three aquifers analyzed. The Kruskal-Wallis test was performed to determine whether there was a significant difference in groundwater composition between the studied aquifers. This method compares numerous independent random samples and can be used as a nonparametric alternative to one-way ANOVA.
Comment 2: Secondly, there are types of this parameters were the “Groundwater Types, Hydrogeochemical Processes, Evaporation, low System Classification Deuterium and Oxígen Isotopic Composition” sectors. Are they including the hydrogeochemical characterization? Response 2: Thank you for your observation. Water types or hydrogeochemical facies and hydrogeochemical processes were considered in the hydrogeochemical characterization. O-18 and Deuterium isotopic signatures were used to identify mainly recharge zones.
Comment 3: Thirdly, in the “Material and methods” section, where are statistical analysis? Thus, I suggest major revisions, and then accepted it. Response 3: We appreciate your suggestion. We include in the methodology the description of the statistical methods used. Page 6, lines 198-206
|
4. Response to Comments on the Quality of English Language |
Point 1: Minor editing of English language required |
|
Once again, thanks a lot for your review. Your guidance will undoubtedly contribute to improving the overall quality of our manuscript.

Reviewer 3 Report
Comments and Suggestions for Authors
General Evaluation:
The current manuscript entitled “Hydrogeochemical Characterization of Groundwater at the Boundaries of Three Aquifers in Central México” by Ibarra-Olivares et al. deals with natural hydrogeochemical mechanisms that govern groundwater chemistry at the margins of the Silao-Romita, Valle de León, and La Muralla aquifers in Mexico's “Bajio Guanajuatense.” This manuscript needs some essential technical and English improvements before finally considering publication in the Water MDPI journal. My specific comments are:
Abstract:
The abstract lacks major numerical findings.
Better to write chemical formulas instead of naming them.
Introduction:
The introduction should provide a clearer rationale for why this hydrogeochemical study is significant, and it should better outline the research objectives.
Correct spacing and syntax problems in the entire manuscript.
Line 49: Supporting reference is necessary.
The hypothesis of the study is missing and a clear linking of problem to solution is poor. Please highlight this before the objectives.
Methods:
Line 71/90: Please provide a range of geocoordinate and agro-climatic information about the study area with appropriate references.
Figure 1: Add north arrow and scale symbols.
The sampling and data collection procedures need to be thoroughly described, including the selection of sampling locations and frequencies. Provide information on how the data was quality-controlled.
Line 161: briefly write about the method of water sample analysis using ion chromatography and support with relevant sources. What was the model and company of the instrument? What about certified reagent materials and quality control measures?
Results and discussion:
The terms and symbols used in data tables are not defined in their footers. Also, coordinate information should be formatted in this way: 21°75’94.40’’ N and 23°03’127.40’’ E. What is the meaning of X and Y, it is not clear.
It is better to conduct a mean comparison test for parameters analyzed at different sampling IDs.
There is a lack of discussion on potential sources of bias or limitations in the sampling and analytical methods. Address this issue to ensure the study's credibility.
The manuscript should provide more context and references related to the three aquifers in Central México being studied, as well as their importance and previous research in the region.
Provide a comprehensive discussion of the results, focusing on the implications of the hydrogeochemical parameters and their relevance to water quality and resource management in the region.
Address the temporal variability of hydrogeochemical parameters, especially if data was collected over an extended period, as this could provide valuable insights.
Please check this issue in the entire manuscript: (Error! Reference source not found.
Conclusion:
Fine.
Comments on the Quality of English LanguagePlease consult a native English speaker or editing service to improve language of your manuscript.
Author Response
Dear Editor and reviewer, I trust this letter finds you well. I wish to extend my sincere appreciation for the time and expertise you invested in evaluating our manuscript titled " Hydrogeochemical Characterization of Groundwater at the Boundaries of Three Aquifers in Central México " submitted to the journal Water, Special Issue "Application of Integrated Geophysical, Hydrogeological and Geospatial Approach to Groundwater Exploration and Contamination". Your thorough assessment and insightful comments have been immensely valuable in refining our work and ensuring its scholarly rigor. Furthermore, your recognition and encouragement inspire us to improve our manuscript even further. We have carefully considered each of your suggestions and critiques, and I am pleased to provide our detailed responses below:
Response to Reviewer 3 |
|
3. Point-by-point response to Comments and Suggestions for Authors |
|
Abstract: Comment 1: The abstract lacks major numerical findings. Response 1: We appreciate your comments. We have added more numerical data on ranges and supplemented it with more information. (see page 1, lines 18 a 22)
Comment 2: Better to write chemical formulas instead of naming them. Response 2: Thank you for your comment, Water family names were replaced by formulas (symbols). (see page 1, lines 22)
|
|
Introduction: |
|
Comment 3: The introduction should provide a clearer rationale for why this hydrogeochemical study is significant and it should better outline the research objectives. Response 3: The hypothesis was added prior to the description of the main objective to improve clarity, as mentioned, and suggested by the reviewer. ( see page 2, lines 63 a 65)
|
|
Comment 4: Correct spacing and syntax problems in the entire manuscript. Response 4: Spacing and syntax problems were corrected throughout the document.
Comment 5: Line 49: Supporting reference is necessary. Response 5: Thank you for your comment. Reference inserted in the text (see Page. 2 , line 52).
Comment 6: The hypothesis of the study is missing and a clear linking of problem to solution is poor. Please highlight this before the objectives. Response 6: Your comment is appreciated. The hypothesis was added before the objectives (see page 2, line 63 to 65).
Methods: Comment 7: Line 71/90: Please provide a range of geocoordinate and agro-climatic information about the study area with appropriate references. Response 7: The coordinates of the study area were added to Figure 1 and climate information related to the study area (see page 2, line 95-111).
Comment 8: Figure 1. Add north arrow and scale symbols. Response 7: thank you for your observation. The north and the graphic scale were added (see Figures 1 and 2).
Comment 8: The sampling and data collection procedures need to be thoroughly described, including the selection of sampling locations and frequencies. Provide information on how the data was quality-controlled. Response 8: The sampling and field data collection procedure is briefly described, according to NOM-014-SSA1, 1993 (page 4, lines 165-178).
Comment 9: Line 161. Briefly write about the method of water sample analysis using ion chromatography and support with relevant sources. What was the model and company of the instrument? What about certified reagent materials and quality control measures?
Response 9: Thanks for your observation, we include information on the quality of the analyses performed. The anions in water were determined by ion chromatography (ASTM D4327-17). Method accredited by the Mexican Accreditation Entity (EMA) with Accreditation No. Q-0401-066/12, effective as of 2022-03-11. Certified reference materials were used (IC-FF-100, IC-CL-100, IC-N, IC-FAS-1A, Anions 698).
Results and discussion: Comment 10: The terms and symbols used in data tables are not defined in their footers. Also, coordinate information should be formatted in this way: 21° 75´94.40”N and 23° 03´127.40” E. What is the meaning of X and Y, it is not clear. Response 10: The symbols used in the physicochemical parameter tables were defined, and the UTM coordinates were changed to geographic coordinates.
Comment 11: It is better to conduct a mean comparison test for parameters analyzed at different sampling IDs. Response 11: Thanks for your suggestion. We performed an analysis of a non-parametric test (Kruskal-Wallis) to determine whether there was a significant difference in groundwater composition between the studied aquifers.
Comment 12: There is a lack of discussion on potential sources of bias or limitations in the sampling and analytical methods. Address this issue to ensure the study´s credibility. Response 12: Thank you for your comments. More information on the sampling methods and quality standards followed in the analysis is included. A statistical analysis of the data was carried out to complement and to know the statistical significance of the data and comparison tests were performed.
Comment 13: The manuscript should provide more context and references related to the three aquifers in Central México being studied, as well as their importance and previous research in the region. Response 13: Thank you for your suggestion. Two paragraphs are added with information on aquifers (see lines 135-148).
Comment 14: Provide a comprehensive discussion of the results, focusing on the implications of the hydrogeochemical parameters and their relevance to water quality and resource management in the region.
Response 14: Thank you for your comment. We have added a statistical analysis of variance comparison to support the differences of the studied aquifers and, at the same time, to strengthen the discussion.
|
|
|
|
4. Response to Comments on the Quality of English Language |
|
|
|
|
Once again, thanks a lot for your review. Your guidance will undoubtedly contribute to improving the overall quality of our manuscript.

Round 2
Reviewer 1 Report
Comments and Suggestions for Authors
The authors have substantially corrected the errors from the previous version of the article. However, there are still a few minor errors that need to be corrected.
Figure 1 - the north mark is still missing
Lines 182-184 – put the text in the chapter on the application of statistical methods.
The text contains errors in the Spanish language in several places
Figure 3 - the resolution of the image is very low, the text of the legend cannot be read
The name of Figure 4 is not visible well
In Figure 8f, part of the text is not visible
Figure 9 - reduce the size of the picture, it is too big and the picture resolution is too low
Figure 10a - reduce the size of the fonts and the size of the symbols for the wells, because the symbols overlap
Lines 502-504 - expand the conclusion on the results of the statistical analysis that determines a certain difference. It is not clear exactly what difference is meant.
In the conclusion, it is necessary to indicate how the results of all analyzes confirmed or refuted the initial hypothesis of the work.
Comments on the Quality of English Language-
Author Response
Dear Reviewer
I would like to thank you for the very constructive comments on our manuscript entitled “Hydrogeochemical Characterization of Groundwater at the Boundaries of Three Aquifers in Central México” (water-2678154). My co-author and I have revised the manuscript to address all your comments and concerns. The revised text in the manuscript is marked in blue.
We feel these changes have greatly improved the manuscript. Please find attached the response to each of your comments and issues raised.
Comment 1: Figure 1 - the north mark is still missing Response 1: Thank you for your observation, the north in Figure 1 has been corrected.
Comment 2: Lines 182-184 – put the text in the chapter on the application of statistical methods. Response 2: Thank you for your suggestion. We include the text in the statistical analysis section on lines 453-459.
Comment 3: The text contains errors in the Spanish language in several places Response 3: We corrected the errors of words in Spanish in the text.
Comment 4: Figure 3 - the resolution of the image is very low, the text of the legend cannot be read. Response 4: Thank you for your comment; the resolution of Figure 3 has been improved.
Comment 5: The name of Figure 4 is not visible well Response 5: Thank you for your observation, the resolution of Figure 4 has been improved.
Comment 6: In Figure 8f, part of the text is not visible Response 6: Thank you for your observation, the resolution of Figure 8f has been improved.
Comment 7: Figure 9 - reduce the size of the picture, it is too big and the picture resolution is too low Response 7: Thank you for your observation; the resolution and size of Figure 9. have been improved.
Comment 8: Figure 10a - reduce the size of the fonts and the size of the symbols for the wells, because the symbols overlap Response 8: Thank you for your observation; the size of the fonts in Figure 10a. has been improved.
Comment 9: Lines 502-504 - expand the conclusion on the results of the statistical analysis that determines a certain difference. It is not clear exactly what difference is meant. Response 9: We include in the conclusions a paragraph explaining the differences between the aquifers studied, supported by the statistical analysis of variance.
Comment 10: In the conclusion, it is necessary to indicate how the results of all analyzes confirmed or refuted the initial hypothesis of the work Response 10: Thank you for your comment. The research's hypothesis is confirmed in the conclusions.
|
Once again, thanks a lot for your review. Without a doubt, your advice will help our manuscript's overall quality.

Reviewer 2 Report
Comments and Suggestions for Authors
The authors give the right answer to the comments.
Comments on the Quality of English LanguageMinor editing of English language required
Author Response
Dear Reviewer
I would like to thank you for the very constructive comments on our manuscript entitled “Hydrogeochemical Characterization of Groundwater at the Boundaries of Three Aquifers in Central México” (water-2678154). My co-author and I have revised the manuscript to address all your comments and concerns. The revised text in the manuscript is marked in blue.
We feel these changes have greatly improved the manuscript. Again, I am extremely grateful for your review. Your advice will undoubtedly improve the overall quality of our manuscript.
Sincerely,
Dr. Raúl Miranda Avilés
Departamento de Ingenierías en Minas, Metalurgia y Geología
División de Ingenierías,
Universidad de Guanajuato, Guanajuato México
